# Inferring the depth and magnitude of pre-instrumental earthquakes from intensity attenuation curves

Paola Sbarra[1], Pierfrancesco Burrato[1], Valerio De Rubeis[1], Patrizia Tosi[1], Gianluca Valensise[1], Roberto Vallone[1], and Paola Vannoli[1]

[1]Istituto Nazionale di Geofisica e Vulcanologia (INGV), Via di Vigna Murata 605, 00143 Rome (RM), Italy

**Correspondence:** Paola Sbarra (paola.sbarra@ingv.it)

**Abstract.** The Italian historical earthquake record is among the richest worldwide; as such it allows the development of advanced techniques for retrieving quantitative information by calibration with recent earthquakes. Building on a pilot elaboration of northern Italy earthquakes, we developed a procedure for determining the hypocentral depth of all Italian earthquakes from macroseismic intensity data alone. In a second step the procedure calculates their magnitude, taking into account the inferred depth.

Hypocentral depth exhibits a substantial variability countrywide, but so far received little attention: pre-instrumental earthquakes were routinely "flattened" at upper crustal level ($\sim 10$ km), on the grounds that the calculation of hypocentral depth is heavily dependent on the largely unknown local propagation properties.

We gathered a learning set of 42 earthquakes documented by reliable instrumental data and by numerous macroseismic intensity observations. We observe (1) that within 50 km from the epicenter the ground motion attenuation rate is primarily controlled by hypocentral depth and largely independent of magnitude; (2) that within this distance the fluctuations of crustal attenuation properties are negligible countrywide; and (3) that knowing both the depth and the expected epicentral intensity makes it possible to estimate a reliable magnitude.

# 1 Introduction

In addition to earthquake magnitude, the severity of seismic ground shaking at any given site is primarily controlled by its geometric spreading, by elastic and anelastic attenuation of the upper crustal rocks, and by hypocentral distance, i.e. the combination of horizontal distance from the epicenter and earthquake depth. Other parameters controlling the ground shaking include the earthquake radiation pattern, the rupture directivity, if any, and the inevitable site amplification effects.

When dealing with damaging instrumental earthquakes, the magnitude, depth and focal mechanism – which in its turn determines the radiation pattern – are generally known, and even the rupture directivity may be at least hypothesized if the recording network is dense enough. Things change drastically when dealing with historical earthquakes. For the vast majority of these events the severity of shaking is expressed by the macroseismic intensity reported at a number of sites, a proxy of a set of accelerometric records (Worden et al., 2012); for all the other parameters we can only make "informed inferences".

Nevertheless, given the limited length of the available instrumental record, historical earthquakes are the primary source of information for the assessment of seismic hazard, at any scale and with any approach. Historical catalogues are especially relevant for assessing seismic hazard in Italy (e.g., Meletti et al., 2021); a country where average recurrence intervals for damaging earthquakes generated by individual sources are very long compared with the length of the instrumental record (e.g., Galli, 2020), but where the historical record of the effects of strong ground shaking is extraordinarily long, spanning over a millennium (Guidoboni et al., 2019). For all of these reasons, it is crucial to establish what information can be actually derived from intensity patterns and how reliable this information is.

Italy affords a unique opportunity to explore what type of information can be realistically derived from intensity data. In the early 1990's macroseismic intensity data started being organized into *analytical historical catalogues* (see the Catalogue of Strong Italian Earthquakes, or CFTI: Boschi et al., 1995, 2000; Guidoboni et al., 2019, 2018), i.e. computer databases where all available individual intensity reports were stored in an orderly fashion, ready to be used for automatic and reproducible elaboration. Later on, the implementation of efficient, Internet-based data acquisition platforms has allowed the systematic investigation of intensity observations also at weak-motion levels, opening new avenues in the interpretation of seismic waves propagation and site response. Over the past 15 years, the platform *Hai Sentito Il Terremoto* (HSIT, Tosi et al., 2007; Sbarra et al., 2019b; De Rubeis et al., 2019) has collected over 1,290,000 felt reports supplied by ordinary people for Italian earthquakes of any size, mostly for weak motions.

Starting at the end of the 1990's, and following the inception of analytical historical catalogues, different workers developed computer algorithms for calculating the earthquake location, its magnitude, and even the presumed rupture orientation and length, for many well-documented pre-instrumental earthquakes (e.g., Musson, 1996; Bakun and Wentworth, 1997; Gasperini et al., 1999; Bakun et al., 2003; Bakun and Scotti, 2006; Sirovich et al., 2013).

Unlike instrumental data, which offer a variety of relevant independent observations (arrival times, amplitudes, phase delays), historical earthquake data are essentially *mono-variable*, meaning that all seismological parameters must be inferred from the same observations: the earthquake macroseismic intensities. Nevertheless, the spatial variation of intensities allows some of the source parameters to be derived. Within this line of research, Sbarra et al. (2019a) proposed a method for estimating the depth

of pre-instrumental earthquakes of northern Italy, whereas other workers (e.g., Valensise et al., 2020) explored the possibility of inferring from intensity data also an indication of rupture directivity.

Aside from the inevitable uncertainties that may arise from such a limited and often poorly distributed dataset, the *mono-variable* nature of the data inevitably leads to the existence of a trade-off among magnitude and depth, because a deeper earthquake will generate smaller shaking, and may thus simply appear as a smaller event. In most cases the magnitude has been estimated without considering the depth, or by fixing it in advance. Other methods were based on a joint inversion of intensity data to obtain magnitude and depth (Traversa et al., 2018; Provost and Scotti, 2020). In any case, depth affects the
observed macroseismic intensity and thus the magnitude estimation of any earthquake (Jánosi, 1906; Kövesligethy, 1907; Blake, 1941; Sponheuer, 1960; Ambraseys, 1985; Burton et al., 1985; Levret et al., 1996; Musson, 1996).

Gasperini et al. (1999) and Gasperini (2001) were certainly aware of these trade-offs (for example, see the discussion in Appendix 2 of Gasperini et al., 1999), but chose to take no countermeasure to minimize their impact. And since the *learning set* used for calibrating their method included almost exclusively instrumental earthquakes that ruptured within the shallowest
portion of the crust, the magnitude values supplied by their method are accurate only for earthquakes occurring in that specific depth interval. Given the large variability of earthquake depth —- in Italy as much as elsewhere — determining a reliable magnitude requires that earthquake depth be properly taken into account, especially in the case of lower crustal or subcrustal events.

At least in Italy, the limited consideration of the depth variability of damaging crustal earthquakes (in this work we are
not concerned with subduction zone events) has often been explained with the inherent difficulty in evaluating the depth of historical earthquakes, motivated by an allegedly large variability in the propagation characteristics of the upper crust (e.g. Mele et al., 1997). Due to the known trade-off between earthquake depth and the properties of seismic wave propagation, this viewpoint — and the resulting decision to fix the depth of historical earthquakes — led the natural variability of earthquake depth to be mapped in terms of variability of crustal properties. For all of these reasons, it is important to use a method capable
of estimating depth and magnitude separately.

Building on the findings of Sbarra et al. (2019a), in this work we characterize the depth and the magnitude of Italian pre-instrumental earthquakes. In particular:

- first we extend the experimental method put forward by these investigators to the whole Italian territory and to the whole pre-1984 earthquake catalogue (CPTI15 v 2.0, the Parametric Catalogue of Italian Earthquakes, Rovida et al.,
2019, 2020). This method was shown to be independent of magnitude, meaning that the *steepness* of the attenuation curve calculated within 50 km (+5 km buffer) from the epicenter is not affected by earthquake size, but only by earthquake depth;

- we then develop a scheme for ranking objectively the quality of an intensity dataset, and hence for selecting only earthquakes that are suitable for calculating a reliable source depth;

- similarly to what was done by Sbarra et al. (2019a) for a northern Italy dataset, we derive equations describing the steepness of the attenuation curve vs. earthquake focal depth from a *learning set*, i.e. a set of relatively recent Italian

earthquakes for which both instrumental and macroseismic observations is available. We then use these equations to estimate the depth of the pre-instrumental events comprising our *analyzed set*;

– finally, from the *learning set* data we derive a multiple regression equation relating expected epicentral intensity to magnitude and hypocentral depth, so as to estimate also the magnitude of pre-instrumental earthquakes.

Notice that the approach we adopted in this work was specifically designed for analyzing also larger magnitude earthquakes ($M_w \geq 6.75$), based on the awareness that

- their causative fault cannot be assumed to be a point source,

- they are often characterized by sizable directivity effects, and

- based on empirical relationships (e.g., Wells and Coppersmith, 1994), the 55 km maximum distance we adopted in our analysis is comparable to the expected length of the causative source of such larger events;

In the process we aim to (a) use our *learning set* to evaluate the properties of wave propagation (within 50 km of the epicenter) in the crust versus the variability of source depth, exploring the trade-off between these two parameters in different tectonic settings; and (b) discuss the potential implications of these developments for the estimation of seismic hazard. The inferred distribution of earthquake depth may have important seismotectonic implications, but these are beyond the scopes of this work and will be discussed in a further, dedicated paper.

## 2   Seismotectonic complexity and depth variability of Italian earthquakes

The Italian peninsula is located along the complex Africa-Europe convergent plate boundary. Due to this complexity, the causative sources of Italian earthquakes exhibit highly variable kinematics and geometrical parameters, as shown by focal mechanisms (Pondrelli et al., 2020) and active stress indicators (Italian Present-day Stress Indicators, IPSI database, Mariucci and Montone, 2020), and as summarized by the Database of Individual Seismogenic Sources (DISS) (Basili et al., 2008; DISS Working Group, 2021). More specifically:

– normal faulting dominates along the hinge of the Apennines chain and in the Calabrian Arc;

– thrust and reverse faulting is widespread along the external fronts of the Southern Alps and of the northern and central Apennines, in the northern and southern Tyrrhenian domain and in the Sicilian-Maghrebian Chain; and

– strike-slip faulting is found in northeastern Italy, in the most external portions of the central and southern Apennines and in the corresponding foreland areas (Figure. 1).

In addition, an active slab related to the subduction of the Ionian lithosphere exists below the Calabrian Arc: the slab is bounded by tear faults along its edges (Maesano et al., 2017, 2020).

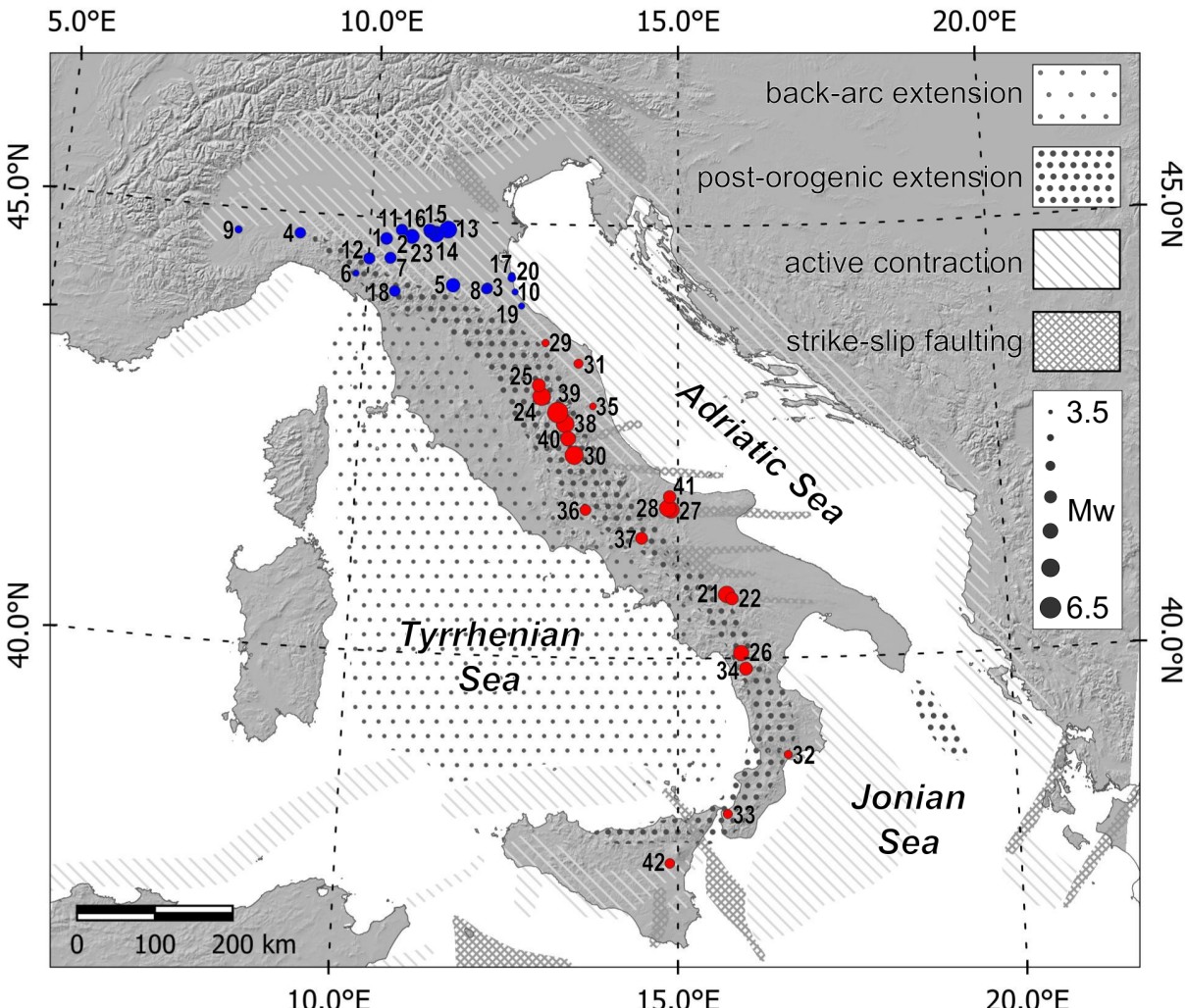

**Figure 1.** Location of the 42 earthquakes of the learning set used in this work and regional-scale tectonic information from the DISS database (DISS Working Group, 2021). The areas with different patterns indicate active tectonic domains that exist in the Italian peninsula and surrounding areas (from Vannoli et al., 2021). The *learning set* includes earthquakes that occurred in northern Italy, shown in blue (Table 1, IDs from 1 to 20 and 23), and in central and southern Italy, shown in red (Table 1, IDs from 21 to 42, except for 23).

The active faults and seismogenic sources identified so far in the Italian region belong both to extensional or compressional fault systems that formed during the presently active stress regime ("*new faults*"), or to structures that formed during previous tectonic phases and were later reactivated with different kinematics ("*inherited faults*"). While the *new faults* cut through the Alps and Apennines fold-and-thrust belts at relatively shallow depth, and are the expression of the ongoing compressional activity or extension due to back-arc stretching or ridge-top collapse, the *inherited faults* are generally rooted at deeper depth
into the crust of the lower plate and are reactivated mostly with compressional and transcurrent faulting mechanisms. Their

depth generally increases moving from the foreland areas, where they may be exposed at the surface (e.g. the Mattinata and Scicli-Ragusa fault systems in the Adriatic and Hyblean foreland areas; Di Bucci et al., 2010), towards the axes of the chain, following the increasing depth of the regional foreland monocline.

The *inherited faults* have been interpreted either as Mesozoic extensional structures characterizing the African northern passive margin and separating fossil paleogeographic domains (e.g., Scardia et al., 2015), or as long-lived faults of various origin, often perpendicular to the architecture of the more recent thrust belts (Zampieri et al., 2021). In addition to this general rule, the recent 2016 Central Italy earthquake sequence has shown that also some large and older thrust faults occurring close to the extensional hinge of the chain may be negatively reactivated if favorably oriented with respect to the current stress field, thus becoming the causative sources of significant normal faulting earthquakes (Bonini et al., 2019; Buttinelli et al., 2021).

Finally, further evidence of the seismotectonic complexity of the Italian region is supplied by the control exerted by the inherited structural and paleogeographic grain of the African paleomargin, which resulted in the segmentation and differential retreat of independent panels of the "foreland monocline", i.e. of the subducting Adriatic, Ionian and Pelagian lithosphere (Mariotti and Doglioni, 2000; Scrocca et al., 2007). As a result of this process, major discontinuities perpendicular to the main structural trends of the Apennines fold-and-thrust belt developed at the boundaries of different foreland monocline panels (e.g., Rosenbaum and Piana Agostinetti, 2015; Vannoli et al., 2015); these discontinuities are highlighted by alignments of geofluid emissions and earthquake swarms (Vannoli et al., 2021), often characterized by transcurrent mechanisms and generally located at deeper depth with respect to the *new faults*, either extensional or compressional.

As a result of this framework, Italian earthquakes exhibit an unusually broad depth range, mainly as a function of their faulting mechanism and of their location in the upper or lower plate (e.g., Chiarabba and De Gori, 2016). They can be grouped in at least four independent depth classes:

- very shallow in the active volcanic areas of the Perityrrhenian margin and of Sicily ($\leq 5$ km);

- shallow ($< 15$ km depth) in both the internal and external domains of the orogen;

- shallow-intermediate ($> 15$ km) in the foreland areas and along large lithospheric tears cutting through the Adriatic monocline and the Apennines (Vannoli et al., 2015);

- deep (up to 600 km depth) in the subduction system below the Calabrian Arc (Chiarabba et al., 2008).

The earthquakes generated by the *new faults* and by the *inherited faults* are often geographically overlapped, as seen in the Po Plain (Sbarra et al., 2019a), which makes their seismotectonic interpretation rather difficult if only the epicentral location is available. Conversely, assigning each pre-instrumental earthquake to a specific depth class helps assigning that event to its relevant domain, thus greatly supporting its seismotectonic interpretation and the calculation of accurate global earthquake budgets and rates.

## 3 Methodology and data analysis

We updated and extended the method proposed in Sbarra et al. (2019a) to make it suitable for use on earthquakes of the entire Italian peninsula through an automated procedure.

We adopt a distance binning method and we use only well-located instrumental earthquakes (see section 3.1). We first calculate the intensity average of individual Macroseismic Data Points (MDPs) falling within 10 km-wide ring-shaped moving windows, so as to obtain an intensity attenuation curve interpolating the resulting 10 intensity averaged points. In most cases the trend of this curve shows an abrupt drop in the attenuation beyond an epicentral distance of about 50 km, as described in Fah and Panza (1994) and Gasperini (2001) and empirically observed by Sbarra et al. (2019a) for earthquake in Northern Italy. For this reason, we calculate the *steepness* of the line that best fits only the first 50 km of the attenuation curve.

A furter issue concerns the treatment of intensity data as integers or real numbers. When estimating macroseismic intensity, all potential diagnostic effects are jointly evaluated to assess which degree of the scale best matches those effects. Typically, however, the effects may belong to contiguous degrees: this circumstance results from multiple reasons, including the geological nature of the outcropping lithology near building foundations, differences in the vulnerability of adjacent buildings, or — for the lowest shaking levels — differences in the perception of seismic vibration depending on the number of stories comprising the building, on whether the observer is still or is moving, and so on (Sbarra et al., 2012, 2014; Oliveira and Ferreira, 2021). Estimation of the shaking effects is even more uncertain for older earthquakes and when only few historical sources are available. The resulting macroseismic intensity is an integer, although the half-degree is often used even in direct field surveys in case of uncertainty between two contiguous degrees. This latter approach implies that intensity values must be processed as real numbers and that an uncertain assessment is either approximated to a half-integer, as proposed by Gasperini (2001), or simply discarded from the data set, as proposed by Albarello and D'Amico (2004). Nevertheless, assigning macroseismic intensities using web-based questionnaires entails greater precision, because it involves using decimal intensities rather than simply integer values (Wald et al., 2006). It has been demonstrated that this procedure leads to lesser scatter than if the calculated intensities were truncated to integers (e.g., Dengler and Dewey, 1998; Dewey et al., 2002). Web based macroseismic data are decimal intensities, while the historical catalogues include half-degree intensities.

In the following step we plot the instrumental depth of the earthquakes used as a *learning set* versus the *steepness* of the attenuation curve. By fitting these values we obtain a logarithmic function that is then used for the last step, that is, to infer the depth of the non-instrumental earthquakes of our *analyzed set*.

Notice that the radius of our ring-shaped moving windows is now calculated from the instrumental earthquake epicenter rather than based on the distance from the epicenter of the innermost MDP within the first 10 km, as proposed by Sbarra et al. (2019a). This minor update makes the algorithm more uniform across the full earthquake magnitude range, and avoids a differential shifting from epicenter for each earthquake (according to the actual distance of the closest MDP from the epicenter). The new approach is summarized in Figure 2 with reference to a specific recent earthquake (20 May 2012; ID 13 of Table 1). After drawing the first 10 km-wide, circular search area centered in the instrumental epicenter (for all *learning set* events), and the following nine 10 km-wide ring-shaped search areas, each one shifted by 5 km from the previous one (0–10, 5–15,

10–20, 15–25, 20–30, 25–35, 30–40, 35–45, 40–50 and 45–55 km: magenta and purple lines in Figure 2b), up to the distance of 55 km from the epicenter, the resulting 10 averaged MDPs intensities are used to build the attenuation diagram (Figure 2c). Subsequently, using the data from the *learning set* we derive a multiple regression equation to calculate the magnitude of the historical earthquakes.

### 3.1 Data selection criteria

To compose our *learning set* (Table 1) we searched the whole Italian territory, selecting all instrumentally well documented earthquakes, i.e. events whose location uncertainties are small and that also feature good quality macroseismic information. For each event the best available source was chosen by expert evaluation based on all available literature sources; Table 1 reports the exact bibliographic source of its depth and magnitude. Whenever a specific study about a given earthquake exists, we used the relocated depth (if available). We built a *learning set* comprising macroseismic data obtained either from a direct survey

or collected through the web, so as to gather information to be used as a sort of "Rosetta stone" for obtaining the parameters of historical earthquakes. For pre-2008 events we used the data stored in the DBMI15 v2.0 catalogue (Locati et al., 2019), a compilation of macroseismic intensities for Italian earthquakes that occurred in the time window 1000 CE-2017, whereas for more recent events of $M_w \leq 5.9$ we used either intensity data from the web-based HSIT catalogue (Tosi et al., 2007; De Rubeis et al., 2019; Sbarra et al., 2019b), or the MDPs collected by targeted post-earthquake surveys conducted by experienced INGV-

QUEST personnel (QUick Earthquake Survey Team; http://www.ingv.it/quest/index.php/rilievi-macrosismici; last accessed on 16 December 2022). Only for the $M_w$ 5.5, 18 January 2017 event we had to make an exception to this rule, due to the incompleteness of HSIT data caused by the evacuations following the $M_w$ 6.0 mainshock of 24 August 2016.

The use of web-based data was fundamental to accomplishing our goals because these data were almost always the only observations available, especially for deeper earthquakes (>30 km). Furthermore, the use of macroseismic data obtained from

200 direct surveys of earthquake damage was fundamental for the correct analysis of the attenuation curves, especially in the epicentral area. The combination of web-based HSIT data and dedicated traditional studies does not affect the results of the *learning set* because the earthquakes that we considered are all relatively recent and their macroseismic field was estimated through a direct field survey. In general, intensity maps drawn for historical earthquakes show more scattered patterns of damage than those revealed by spatially rich, web-based intensity data for similar-sized, recent events (Hough, 2013, 2014).

This problem particularly affects earthquakes whose effects are estimated through written sources. The same is true if only written sources (e.g., newspapers) are used to estimate the intensities of recent earthquakes; they will inevitably end up being overestimated (Sbarra et al., 2010; Hough, 2014).

The events comprising the *learning set* were further selected based on the following criteria:

1. pre-2007 earthquakes must have $M_w \geq 5.0$, but events of $M_w \geq 4.5$ are also accepted if they are backed by a targeted
study;

2. post-2007 earthquakes must have $M_w \geq 4.0$ if their depth is $> 25$ km, or $M_w \geq 4.5$ if their depth is $\leq 25$ km;

3. the earthquake depth must not have been fixed *a priori* by INGV's National Seismic Network;

4. only for pre-2012 earthquakes, the event must not be an aftershock occurring within a week of the mainshock, or a foreshock that occurred less than 24 hours before the mainshock;

5. all earthquakes having $M_w \leq 5.8$ must not be aftershocks of the central Italy sequence of 2016;

6. the earthquake must be documented by at least 100 MDPs, at least 60 of which must fall within the first 55 km from the epicenter;

7. the MDPs falling at a 10–55 km distance from the epicenter must be distributed in an azimuthal range $\geq 180°$;

8. the attenuation *steepness* must be calculated based on six or more averaged points, thus at least 6 of the 10 rings must contain suitable MDPs;

9. the standard error of the estimated attenuation *steepness* must be $\leq 0.01$.

All 42 earthquakes listed in Table 1 fulfill these rather strict criteria with the only exception of #6 and #17 (MDP < 60), two deeper events that were already included in the learning set of Sbarra et al. (2019a) as they are crucial for characterizing lower crustal and subcrustal seismicity.

Notice that the selection criteria 1–4 had already been adopted by Sbarra et al. (2019a). The additional criteria 5 through 9 were added in consideration that the present work deals with the entire Italian territory, and hence with a much larger diversity of the potentially concerned earthquakes. More specifically:

– criterion #5 was added due to the recurring lack of data in the epicentral areas of the main aftershocks of the 2016–2017, central Italy sequence, due to the widespread evacuations following the $M_w$ 6.0 mainshock of 24 August 2016 and to the superposition of the effects of subsequent shocks;

– criterion #6 was added after various experimental tests, in order to achieve more reliable and stable estimates of the attenuation;

– criteria #7 and #8 were introduced to discard earthquakes located offshore or near the coastline, whose epicentral location generally exhibits greater uncertainty;

– criterion #9 was adopted to retain only earthquakes for which we could calculate a reliable attenuation *steepness*.

## 3.2 Analysis of the learning set

We analyzed separately two data subsets, respectively comprising only earthquakes located in Northern Italy, and earthquakes located in the rest of the Italian peninsula. We made this choice because the dataset used in Sbarra et al. (2019a) included only earthquakes from a region whose lithospheric structure and wave propagation properties are homogeneous (Mele et al., 1997; Gasperini, 2001). Conversely, in this work we wanted to evaluate the possible influence of variable attenuation properties resulting from the full range of tectonic and geodynamic diversity characterizing the Italian peninsula.

Due to the intervening minor updates in our methodology — and specifically in the calculation of the starting point of our moving window, which implies a slightly different *steepness* for the first 50 km of the attenuation curve (Mean= -0.00015; Max= 0.007) — we first recalculated the attenuation *steepness* for all the 20 earthquakes comprising the *learning set* used by Sbarra et al. (2019a) (Table 1, ID from 1 to 20; see Figures 1 and 3). We added to this dataset the 1996, Emilia earthquake (ID 23), originally rejected because its depth from the ISIDe catalogue (ISIDe Working Group, 2007) was fixed at 10 km; for this event we now use the depth evaluated by Selvaggi et al. (2001). We then analyzed the earthquakes we selected for the rest of Italy and calculated their intensity attenuation *steepness* (Figures 1, 4; Table 1, from #21 to #42, except for #23).

As discussed earlier, in both datasets, which together form our new *learning set*, we observed a distinct break in *steepness* at an epicentral distance of about 50 km (see Figures 3, 4,). In describing this feature of the Italian attenuation curves, Gasperini (2001) contended that within a $\sim$ 50 km epicentral distance the ground shaking is dominated by direct seismic phases, whose propagation is highly sensitive to earthquake depth, whereas Moho-reflected phases dominate at larger distances. According to this hypothesis, the exact distance of the transition would be controlled by the average Moho depth along the source-receiver path.

For all the earthquakes of the *learning set* we then plotted the *steepness* (S; intensity/km), i.e. the absolute value of the attenuation slope, versus focal depth (D; km), and found two separate but very similar best-fitting logarithmic functions (Figure 5). For northern Italy we found

$$S = (-0.020 \pm 0.006)\ln D + 0.093 \pm 0.018 \tag{1}$$

whereas for central and southern Italy we found

$$S = (-0.016 \pm 0.007)\ln D + 0.079 \pm 0.019 \tag{2}$$

The coefficients of each function fall within the 95% confidence interval of the the other function, suggesting that our method does not detect any statistically significant change in the attenuation of macroseismic intensity between the two domains, at least over the first 50 km of epicentral distance. This finding also suggests that an approach based on averaging the intensity values distributed over circular search areas has the ability to smooth out most of the inevitable azimuthal differences in crustal propagation properties.

We decided to calculate a new logarithmic function using all 42 earthquakes of the *learning set*, so as to obtain a law that may be used over the whole Italian region (green line in Figure 5):

$$S = (-0.018 \pm 0.004)\ln D + 0.087 \pm 0.013 \tag{3}$$

The regression F-test of the three regressions is acceptable at a significant probability level $P < 0.0001$. As we expected, Eq. 3 is similar to the previous two equations, and exhibits narrower 95% confidence bands (Davis and Sampson, 2002)

resulting from the larger number of available data points. Eq. 3 can be applied for a *steepness* interval $0.058 \leq S \leq 0.010$, which corresponds to a depth interval $5 \leq D \leq 73$ km. Notice that the function is not constrained beyond these limits, and hence should not be used for shallower or deeper events. For depths greater than 35 km and steepness less than 0.02, the uncertainty is larger. Consequently, the confidence bands of Eq. 3 in Figure 5 exhibits wider limits, yet they still provide valuable information on depth estimation, albeit within a wider error range. Notice also that for epicentral distances $> 50$ km the curves shown in Figures 3 and 4 exhibit slightly different steepness between the northern and the central-southern Italy datasets (respectively, 0.01 and 0.02), in agreement with the observations made by Gasperini (2001). Conversely, as mentioned above, in the first 50 km of the attenuation curve there does not seem to be any influence by crustal attenuation properties, hence in this case a trade-off exists only between magnitude and depth. In contrast, the intensity attenuation for epicentral distances >50 km earthquakes occurring in northern Italy, where the crust is generally thicker than in the rest of the country, frequently shows a characteristic, very gently sloping plateau that has been interpreted as due to Moho reflected seismic waves by Bragato et al. (2011). A further element to be taken into account is the difference in seismic wave propagation between the Tyrrhenian and Adriatic sides of Italy, most likely resulting from a rather different efficiency of the seismic energy propagation of the crust-upper mantle system (Mele et al., 1997; Lolli et al., 2015; De Rubeis et al., 2016; Di Bona, 2016). But again, no differences are appreciated in our analysis for earthquakes located on the Tyrrhenian or Adriatic sides with epicentral distances up to 50 km (see Figures 1, 3 and 4).

### 3.3   Independence of the method to infer the earthquake depth from magnitude

The *steepness* of the first 50 km of the attenuation curves calculated for the earthquakes of our *learning set* (Figures 3, 4) is independent from magnitude, as already empirically observed by Sbarra et al. (2019a) for a smaller sample of events. To prove this statement we correlated the *steepness* with $M_w$ in addition to the natural logarithm of depth (see Eq. 3) and found that its coefficients are not significant (95% confidence interval includes the null value). As an example of the independence of the *steepness* from magnitude we plotted in Figure 6a the attenuation curves for four earthquakes falling in the rather wide $M_w$ range 4.8–6.5, but having a similar instrumental depth, in the range 7.3–8.7 km (#4, 16, 24, 39; see Table 1). Figure 6a shows that all the calculated *steepness* values fall in a rather narrow range (0.045–0.051), regardless of magnitude. Figure 6b shows that the same behavior is observed also for four deeper earthquakes (#8, 10, 11, 26; see Table 1), which share a similar instrumental depth (24.5–29.2 km) but exhibit a different $M_w$ (4.0–5.6).

The invariance of the attenuation steepness with magnitude for *learning set* events is a key point as it makes our approach suitable for analyzing historical earthquakes even if their size is not well constrained. Instead, other methodologies (Traversa et al., 2018; Provost and Scotti, 2020) are subject to a trade-off between depth and magnitude, as both parameters are treated as unknown. Our approach is similar to that of previous investigators (Jánosi, 1906; Kövesligethy, 1907; Blake, 1941; Sponheuer, 1960; Burton et al., 1985; Musson, 1996), who based their analyses on manually drawn isoseismals, but directly uses the fit of the attenuation curve computed on averages of the original MDP falling inside moving circular windows.

### 3.4 Comparison with synthetic models

We analyzed the possibility of reproducing the empirical trend of Figure 5 through predictive models, expressing the macro-
seismic intensity (Intensity Prediction Equations, IPE) or the Peak Ground Acceleration (PGA) as a function of the earthquake
magnitude and distance. It is worth noting that many of the IPEs and GMPEs (Ground Motion Prediction Equations) proposed
in the literature (e.g., Douglas, 2003, 2021) assume a predetermined depth for all earthquakes considered. The difficulty in
considering this parameter is due to the uncertainty associated with the depth itself, not only for historical earthquakes but also
for recent events located in areas that are geologically complex, or not monitored by a dense seismic network. To explore the
variation of the attenuation steepness with depth we therefore used three of the models that explicitly include this parameter: the
IPE by Tosi et al. (2015), the IPE by Musson (2013), and the GMPE by Cauzzi and Faccioli (2008). We chose these functions
because they feature a simple functional form which determines a magnitude-independent attenuation steepness, as suggested
by real earthquake data. Conversely, a functional form containing a term combining magnitude and distance would lead to a
change in the shape of the attenuation curve with distance and to a variation of the *steepness* for a variable magnitude.

We then used two different conversion equations (Faenza and Michelini, 2010; Masi et al., 2020) to convert the PGA values
obtained with the adopted GMPEs into MCS macroseismic intensities, so as to test also the influence of the conversion process.
We used these equations to compute the macroseismic intensities caused at several epicentral distances by a hypothetical $M_w$
5.0 earthquake located at variable depth. We then applied the same 10 km-moving window average method used for the
analyzed earthquakes and calculated the regression line within a distance of 50 km.

Figure 7a shows some of the average intensity values obtained using the IPE proposed by Musson (2013) for a magnitude
$M_w$ 5.0, along with their regression lines. We remark that, although the macroseismic intensity is proportional to the logarithm
of the hypocentral distance, the linear regression of intensity versus epicentral distance gives statistically significant results in
the adopted distance range. In addition we show that, even using an IPE, 50 km is a reasonable limit for a linear regression that
well approximates the first part of the attenuation curve. Figure 7b shows the *steepness* of the regression lines thus calculated as
a function of the earthquake focal depth, and the values, calculated with the same method, derived from PGA using the GMPE
proposed by Cauzzi and Faccioli (2008), converted into macroseismic intensity. It is worth noting that differences caused by the
use of two different conversion equations are greater than the differences caused by the use of the IPE in place of the GMPE.
At any rate, in all three cases the trend of the values as a function of the depth is similar to the trend we found empirically. The
greater difference is observed for depth larger than 35 km, probably because the empirical regression is less constrained for
such deep events. This is reflected in the wider confidence bands of (Eq. 3) (see Figure 5), due to fewer learning-set earthquakes
at those depths. Having been obtained with a completely different kind of data, this result suggests that the approach followed
for deriving Eq. 3 is adequate and reliable.

### 3.5 Reliability and validation of the depth estimation method

The reliability of the *steepness* of the first 50 km of the attenuation curve depends on the quality and spatial distribution of
the available MDPs and on the accuracy of the epicentral locations. Italian macroseismic data are systematically stored in the

DBMI database v4.0 (Locati et al., 2022); as a rule of thumb, the older is the earthquake, the less complete and reliable are the historical sources from which macroseismic intensities were derived (e.g., Guidoboni and Ebel, 2009).

To test our procedure we investigated the minimum number of MDPs of the macroseismic field that are needed to obtain an estimate of the attenuation steepness. To this end we intentionally and randomly depleted the macroseismic field of the 20 May 2012, $M_w$ 5.8, Emilia earthquake (#13), a well-recorded event for which over two hundred spatially well-distributed MDPs are available, using data from the HSIT database (De Rubeis et al., 2019); see Figure 2a. For each of the ten ring-shaped search areas (see Figure 2b, 2c) we performed a gradual reduction of the number of MDPs (from 1% to 99%), the same for each step and for all areas (see Figure 8). For example, let us consider three adjacent ring-shaped areas: the first with 32 MDPs, the second with 18, and the third with 9 MDPs. The depletion procedure would lead (among others) to the following steps: a 35% depletion will leave 21 MDPs in the first area, 12 MDPs in the second, and 6 MDPs in the third; a 68% depletion will leave 10, 6 and 3, respectively; a 97% of depletion will leave 1, 1 and 0 MDPs.

The linear fit of the attenuation trend was calculated 1,000 times for each depletion step, so as to evaluate the *steepness* variability through its standard deviation.

Figure 8 shows the number of MDPs versus the standard deviation of the *steepness*, which is equal to 0.01 when only 14% of the total data are left, corresponding to 30 MDPs; this implies that the most likely *steepness* values (68%) fall within a standard deviation equal to ±0.01.

Our depletion test shows that we may obtain an acceptable attenuation steepness even for historical earthquakes for which at least 30 MDPs are available, provided that they are homogeneously distributed for each distance window. Moreover we calculated the depth reliability by estimating the depths corresponding to the confidence bands eq. 3 for each calculated *steepness* of the *analyzed set* earthquakes (see Figure 5); these values are shown in Table S1. We also calculated the depth of the 42 events in our learning set using the proposed methodology (Table S2, supplementary Figure S1)

### 3.6 A two-step method for estimating magnitude based on intensity and depth

While the inferred hypocentral depth is independent of magnitude and can be obtained simply based on the *steepness* of the line that best fits the first 50 km of the attenuation curve, the estimation of the magnitude itself affects the *y-intercept* (the expected intensity at the epicenter, $I_E$) of the linear fit (Figure 6): for a constant depth the *y-intercept* increases for an increasing magnitude (Figure 6), and decreases if depth increases for a constant magnitude. Therefore, a reliable magnitude determination based on macroseismic data must necessarily take into account earthquake depth.

We devised a two-step procedure where depth is estimated first (Step 1), then $M_w$ is empirically estimated using our *learning set* data to derive a standard least squares regression equation among D (depth), $I_E$ and $M_w$ (Step 2):

$$M_w = (0.18 \pm 0.19)\ln D + (0.56 \pm 0.11)I_E + (1.44 \pm 1.06) \qquad (4)$$

Eq. 4 can be applied for a depth interval $5 \leq D \leq 73$ km and $3.5 \leq I_E \leq 8.1$. Figure 9 shows the *learning set* data used in the regression, together with the magnitude isolines of Eq. 4. This relationship shows the extent of geometric attenuation of intensity due to the propagation of seismic waves from the hypocenter to the epicenter.

For $lnD$ the 95% confidence interval of the coefficient includes the null value, however, the coefficient becomes significant at a slightly smaller confidence level (93%). Magnitudes obtained through this procedure are referred to as *y-intercept* $M_w$. In this perspective the attenuation curve becomes a sort of 'earthquake identity card', as it contains all the elements needed to retrieve magnitude and depth from the observed intensities, provided that a reliable calibration scheme is available. Such calibration can be regarded as an application to Seismology of the *principle of actualism* popularized by British naturalists in the late 18th century: "Observing modern earthquakes to understand those of the past".

Deriving magnitude using only well-studied earthquakes with their expected epicentral intensities provides a better estimate of $M_w$ because it is based on larger intensities than that obtained by inverting an IPE Sbarra et al. (2019a).

The method summarized by Eqs. 3 and 4 is simple and intuitive, plus it may allow a geological verification of the depth before estimating the magnitude. Our Step 2 uses a method similar to that proposed by Gutdeutsch et al. (2002), who applied it only to carefully selected datasets, so as to minimize the bias caused by a poorly constrained depth or by an incomplete macroseismic field.

In conclusion, starting from our empirical observations of the independence of the attenuation steepness from magnitude, we were able to mitigate the trade-off between magnitude and depth when estimating both these parameters from macroseismic data.

## 3.7  Influence of macroseismic cumulative effects on the depth and magnitude estimation

It is hard to estimate macroseismic intensities for individual earthquakes occurring close in time and space (multiple events, strong aftershocks, etc.; e.g. Grünthal, 1998; Grimaz and Malisan, 2017; Graziani et al., 2019), particularly in the case of historical events. Macroseismic data may then reflect accumulated effects, a circumstance that would ultimately affect the attenuation steepness and hence contaminate the inferred earthquake depth and magnitude. This is a recurring problem in historical earthquake catalogues; a condition that is hard to overcome even for modern earthquakes, and even if a very rapid damage survey is carried out, because the first large shock inevitably causes an increase in the vulnerability of the building stock whose effects on later shocks are virtually impossible to identify.

To quantify the effect of multiple events on the determination of earthquake depth we analyzed the 29 May 2012, $M_w$ 5.7, Emilia earthquake, one of the *learning set* events, using the MDPs from DBMI15 instead of those supplied by HSIT (ID 14 in Table 1).For this event the DBMI macroseismic field (Tertulliani et al., 2012, https://emidius.mi.ingv.it/ASMI/event/20120529_0700_000) includes the effects of the 20 May event ($M_w$ 5.9; https://emidius.mi.ingv.it/ASMI/event/20120520_0203_000), which occurred nearby. As expected, these circumstances misled our method, causing a drastic overestimation of the earthquake depth (36.8 km). Conversely, thanks to the rapidity in the response given by citizens and to the ensuing lack of contamination, the HSIT dataset method returned a depth $\leq$ 5 km, much closer to instrumental estimate (8.1 km).

A similar case of contamination could be that of two earthquakes that occurred seven months apart in two distinct but relatively close areas of the northern Apennines; the 10 November 1918, $M_w$ 6.0, Appennino forlivese, and the 29 June 1919, $M_w$ 6.4 Mugello shocks (IDs 101 and 102 in Table S1, respectively). Our work estimates a depth range of 19–27 km for the 1919 earthquake (see Table S1). This result is incompatible with the estimated depth of the DISS seismogenic source that is deemed responsible for the 1919 event (ITIS086; depth range of 1–7 km based on seismotectonic evidence; DISS Working

Group, 2021). We suspect that this anomalous depth estimate (19–27 km) could be explained by a sort of overlap between the two macroseismic fields, because a portion of the intensity pattern of the 1919 earthquake overlaps the region struck by the 1918 shock (Rovida et al., 2021). As a result, the (apparent) intensity pattern of the 1919 earthquake is likely contaminated by the 1918 event. Both in the case of the 2012 sequence and in the case of the 1918-1919 earthquakes, the inferred depth of the second mainshock is deeper than that shown by instrumental data, or suggested by geological observations, due to the overlap

between the two macroseismic fields. The intensity fields for 29 May 2012 and 29 June 1919 appear more spread out than they should, due to the contamination from the previous earthquakes; this entails a lower steepness of the attenuation curve — corresponding to an apparently lower attenuation of macroseismic intensity — which ultimately translates into a deeper depth.

### 3.8   Dealing with larger magnitude earthquakes

Our approach works well if the size of the seismic source is negligible relative to the epicentral distances, but it may not be
immediately applicable to estimate the attenuation of the macroseismic intensity for a large magnitude earthquake (Gasperini, 2001; Albarello and D'Amico, 2004; Pasolini et al., 2008). To test the validity and possible limitations of this assumption we evaluated the maximum magnitude for which the use of a point-source approximation is granted, using both our *learning set* and our *analyzed set*.

    We used the empirical relationships proposed by Wells and Coppersmith (1994) to calculate the rupture area and the expected
length of the seismogenic source based on the *y-intercept* $M_w$ (Eq. 4). Assuming a dip angle of $45°$ for every fault, irrespective of its kinematics and tectonic setting, we calculated the surface projection area of each rectangular source and the radius $R_e$ of the equivalent circle (i.e. a circle having the same projected area as the fault: thus $R_e$ is a function of $M_w$). We found $R_e$ greater than 10 km only for earthquakes of $M_w \geq 6.75$ (10 km is our standard radius of the moving circular search areas); but not having any such earthquakes in the *learning set* (Table 1), we used a geometric correction only to infer the depth of 21
earthquakes of the *analyzed set* (see discussion at the end of this section and in Table S1).

    Then we applied to this group of larger magnitude earthquakes a procedure that we call "variable moving windows". More specifically, we used as the first search area a circular window of radius $R_e$, inside which we averaged the MDPs intensities, while for the subsequent windows — each one shifted by 5 km, as usual — we adopted the standard 10 km radius increase. For the 13 January 1915, $M_w$ 7.1, Marsica (central Apennines) earthquake, one of the largest in Italian history and for which
there exists a very reliable macroseismic dataset, we made a test using the $R_{JB}$ distance following the method implemented by Joyner and Boore (1981) and calculating the average of the MDPs inside a rectangular rather than a ring-shaped area. We singled out this earthquake because it was a single mainshock event and its macroseismic field should not have been contaminated by the effects of previous or later significant events (see section 3.7)

The comparison of the attenuation *steepness* calculated using the $R_{JB}$ distance or using the moving window or variable moving window methodology proposed here shows only modest fluctuations. For the 1915 earthquake we found a *steepness* of 0.044, 0.044 and 0.047, respectively using the $R_{JB}$ approach, the moving window approach and the variable moving window approach. These *steepness* fluctuations imply a difference of about 1.5 km in the expected depth. Given the uncertainty about our knowledge of the source geometry for historical earthquakes and the limited impact of using the $R_{JB}$ distance in the window approach, we decided to recalculate all distances for earthquakes having $M_w \geq 6.75$ using the variable moving windows method only, which analyses the MDPs over circular search areas.

In conclusion, using an extended source approach for the largest earthquakes has a minimal influence on the *steepness*. Conversely, the effect on the *y-intercept* ($I_E$) is not negligible. The correct way of calculating their magnitude would be using $R_{JB}$ distances, but due to the lack of information on source geometry we use the variable moving windows method and apply a geometric correction to the intercept value. As a result, for 21 earthquakes having $M_w \geq 6.75$ (see Table S1) we assumed an extended source with a radius of $R_e$. Consequently, the distances of the relevant MDPs were systematically reduced by $R_e$, leading to a geometric correction of the regression line and of its intercept: $I_E = I_E - S * R_e$ where $S$ is the *steepness*.

Finally, we recalculated the magnitude of these 21 earthquakes using Eq. 4.

### 3.9   Reliability of the magnitude estimation method

Since our is a two-step method and magnitude is calculated after estimating depth, we provided the estimate of the error associated with the magnitude of the *analyzed set* earthquakes, based on the confidence limits of depth, by applying Eq. 4 for the lower and higher depth limit based on 95% confidence band (Table S1).

As a countercheck we used our method to calculate the depth first (Eq. 5) and then the magnitude (Eq. 4) of the 42 events of our *learning set* (Figure 10, Table S2, see also magnitude residuals in supplementary Figure S2), so as to analyze the difference from the instrumental magnitudes listed in Table 1. We obtained differences in the range 0.68 to -0.41 magnitude units (supplementary Figure S2), respectively, with an average of -0.03 and a standard deviation of 0.28.

We then compared the macroseismic magnitudes calculated through our method with those calculated through the Boxer method (Gasperini et al., 1999), using the very same intensity dataset from DBMI15. Notice, however, that the parameters of the earthquakes comprising our *learning set* were computed using also data from other sources, such as HSIT, CFTI5Med etc. (Table 1). Table 2 lists the magnitude of all *learning set* earthquakes for which the comparison was possible. Notwithstanding the significant differences between the two methods, the root mean squared error between instrumental magnitudes and those estimated with Boxer and our method turned out to be comparable; respectively 0.38 and 0.35.

Finally, the reliability of the *y-intercept* $M_w$ is primarily a function of the accuracy of the macroseismic field from which $I_E$ is derived, but also of the estimated depth. For instance, we examined the 13 January 1909, Northern Italy earthquake (https://emidius.mi.ingv.it/ASMI/event/19090113_0045_000), whose macroseismic field is suggestive of a rather deep source. We obtained a depth of 44 km, yielding a $M_w$ 5.5; should this earthquake be much shallower (e.g. 5 km), Eq. 4 would return a substantially smaller $M_w$ (5.1).

## 4   Application to the CPTI15/DBMI15 catalogues

We applied our methodology to the pre-1984 earthquakes of the DBMI15 v2.0 catalogue (Locati et al., 2019). We analyzed only pre-1984 events because their parameters were computed from intensity data as their instrumental location is generally unreliable, although there are notable exceptions (see discussion in Rovida et al. (2021)).

We first selected the earthquakes to be analyzed: they must meet all criteria listed in Sect. 3.1 "Data Selection criteria". We made an exception only for earthquake #6, for which we reduced the minimum number of MDPs from 60 to 30 based on the conclusions drawn in Sect. 3.5 "Reliability and validation of the depth estimation method". These criteria were passed by 206 out of 2,679 earthquakes (Figure 11 and Table S1), which comprise the *analyzed set* of this work. Unfortunately, the vast majority of the events listed in DBMI15 exhibit less than 30 MDPs within the first 55 km from the epicenter, and therefore had to be discarded. To estimate the depth of the 206 events that were retained we first calculated the *steepness* of the line that best fits the first 50 km of the attenuation curve of each event, then we used Eq. 5, which is simply the reverse of Eq. 3:

$$D = e^{\frac{0.087 - S}{0.018}} \tag{5}$$

We recall that D is the depth and S is the *steepness* (see section 3.2 for the application ranges).

In Sect. 3.2 we clarified that we can calculate a reliable depth only for events whose *steepness* falls in the interval 0.058 to 0.012 (Figure 5), which corresponds to the depth interval 5.0–73.0 km, respectively (Table S1). This implies that an inferred 5.0 km depth must be intended as $\leq$ 5.0 km, and similarly, a 73.0 km depth stands for $\geq$ 73.0 km.

We wish to stress once again that the reliability of the magnitude and depth determinations shown in Figure 11 and Table S1 depends on both the quality of the macroseismic data and on the accuracy of the epicentral locations. For completeness of information, Table S1 also reports the full details of the processing for each of the selected events; in addition the supplementary .zip files S1 and S2 contain the histograms of the number of MDPs in ranges of distances up to 50 km from the epicenter (as in Figure 2), and the attenuation curves of the 206 analyzed set earthquakes, thus allowing a detailed examination of all analyzed data. The uncertainty associated with the inferred depths is determined by the confidence bands shown in Figure 5, and is hence larger for deeper earthquakes (Table S1 shows the depth range obtained by calculating the lower and upper limits of the 95% confidence band). In addition, Eq. 3 and Eq. 5 are affected by the accuracy of the instrumental location of the *learning set* earthquakes, on the basis of which the logarithmic curve data are fitted. Some of the inferred depths have larger confidence intervals, due to inherent uncertainties that are reflected in the determination of the *steepness* and of the *y-intercept* (as defined in Sect. 3.8): these may include the cumulation of damage (see section Sect. 3.7) from subsequent shocks, unpredictable anomalies in wave propagation, strong source directivity and site amplification effects, all of which may also cause a sizable shift in the epicentral location.

Once the depth of the 206 selected earthquakes is known, we can estimate their magnitude using Eq 4. All estimated depths and magnitudes are shown in Figure 11 (see Table S1). We used the equations on the analysed set even beyond the application limits of $I_E$ to still estimate an indicative magnitude (in these cases magnitudes are marked in Table S1 with an asterisk).

Before comparing the $M_w$ estimates obtained with our approach and those listed in the CPTI15 catalogue we must recall that all $M_w$ estimates supplied by this catalogue are inherently hybrid. When the catalogue reports both an instrumental and a macroseismic epicenter, the decision to adopt the macroseismic or the instrumental value as 'preferred' is made on a case-by-case basis by the catalogue compilers. For the vast majority of these 206 events CPTI15 adopted the intensity-based magnitude as the preferred $M_w$ (Rovida et al., 2021).

Figure 12 shows a comparison of the magnitude obtained with our methodology with the corresponding magnitude listed in CPTI15 (Table S1). The two sets of estimates are generally consistent, yet on average the magnitudes calculated in the present work show a difference of +0.25 magnitude units. Moreover, Vannucci et al. (2021) stated that the magnitudes of all pre-instrumental earthquakes in CPTI15 might be overestimated by 0.1–0.2 units due to differences in the response of pre-1960 seismographs relative to the response of more recent and better calibrated electromagnetic sensors. Recalibrating the Boxer coefficients for magnitude calculation using only events from 1960 to 2009 results in macroseismic moment magnitudes that are lower than those reported by the CPTI15 by 0.144 magnitude units, on average (Vannucci et al., 2021). If this were the case, the difference between our estimates and the CPTI15 estimates summarized in Figure 12 would be even larger.

The calculated $M_w$ may also vary if we consider macroseismic intensities assigned using the MCS or the EMS scale; according to Vannucci et al. (2021), using one or the other may cause differences in the macroseismic location.

It is important to be aware that the calculation of $M_w$ from macroseismic data, using either Boxer or our methodology, is controlled by a number of variables whose relative weight is critical: assigning proper weights, however, is not an easy task, regardless of the quality of the data and of the reliability of the adopted algorithm.

The +0.25 magnitude unit difference we found implies that on average our seismic moments are 2.3 times larger than those obtained using the Boxer method; a conclusion that may have strong implications for the assessment of seismic hazard.

## 5 Conclusions

In this study we present a two-step procedure for deriving the depth and magnitude of Italian pre-instrumental earthquakes from official, publicly accessible macroseismic intensity datasets: the traditional macroseismic historical dataset supplied by DBMI15, and the new web based macroseismic HSIT dataset. The main merit of the proposed methodology is its objectivity and ease of application.

Web-based macroseismic platforms allow a large amount of data to be collected through crowd-sourcing; they are often the only available source of information concerning the effects of low-to-medium magnitude earthquakes, and of the far-field effects of larger events. In fact, HSIT data were critical to perform this work because — especially for deeper earthquakes ($> 30$ km) — they were almost always the only available macroseismic observations available for our *learning set* .

We proved that the initial 50 km of the attenuation curve contain all the elements needed to retrieve not only the depth, but also the magnitude of any given earthquake. The methodology was tested on Italian earthquakes, but we maintain that it can be extended to other countries, following the necessary calibrations.

The first step of our procedure involves the calculation of earthquake depth (Eq. 5). Based on our empirical observations we show that the *steepness* of the attenuation curve in the first 50 km from the epicenter does not vary much due to regional differences in seismic wave propagation properties (Figure 5), so that for these distances the only significant trade-off is that between depth and magnitude. We also show that, at least in our *learning set*, the *steepness* of the attenuation curve in the first 50 km from the epicenter appears to be independent of magnitude but is largely a function of source depth. This finding implies that the propagation properties do not change much countrywide, despite the well-known complexity of Italian geodynamics and the ensuing geological heterogeneity; as a result, our new relations are valid for the whole Italian territory (Figure 5).

The second step involves estimating the magnitude through an empirical law obtained from a regression function that relates the expected epicentral intensity to the depth and magnitude of the 42 earthquakes comprising our *learning set* (Eq. 4). We applied this methodology to 206 earthquakes from the CPTI15 catalogue, after removing all events whose macroseismic field is too sparse or too inhomogeneous to return reliable results.

Our approach allowed us to verify that the inferred depth is consistent with the presumed earthquake-causative tectonic structures, and is essential to obtain a well-calibrated magnitude value. We contend that the new methodology may be crucial for mitigating the trade-off between earthquake depth and magnitude; this is a pre-condition for calculating reliable depth estimates — and hence reliable magnitudes — for earthquakes of the pre-instrumental era.

In Italy the historical record is still the main pillar of any seismic hazard analysis, conducted at any scale and using any approach. We maintain that the revised framework discussed in this work may ultimately serve for exploiting more systematically the enormous potential of historical earthquake data, and ultimately for providing inherently more reliable input data for seismic hazard assessment.

*Code availability.* Code cannot be shared at this stage

*Data availability.* This work used only published or public domain datasets

*Author contributions.* P.S. conceived the work and wrote the initial draft of the paper, P.B., P.T., P.V. and G.V. contributed to delineating the structure of the paper. P.B., P.V. and G.V. provided information on the seismotectonic background, along with the associated interpretations, P.S. and R.V. analyzed the macroseismic data and R.V. implemented the algorithms in the R language. P.T. tested the method through the use of synthetic data. V.D. evaluated statistically the effects of using finite seismic sources. P.S. and G.V. did most of the writing. All authors discussed the results and contributed to the final version of the paper.

*Competing interests.* The authors declare that they have no conflict of interest.

*Acknowledgements.* We thank Franco Mele, Mario Locati, Graziano Ferrari, Livio Sirovich and Franco Pettenati for suggestions and for providing valuable insight during the early stages of this work.

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

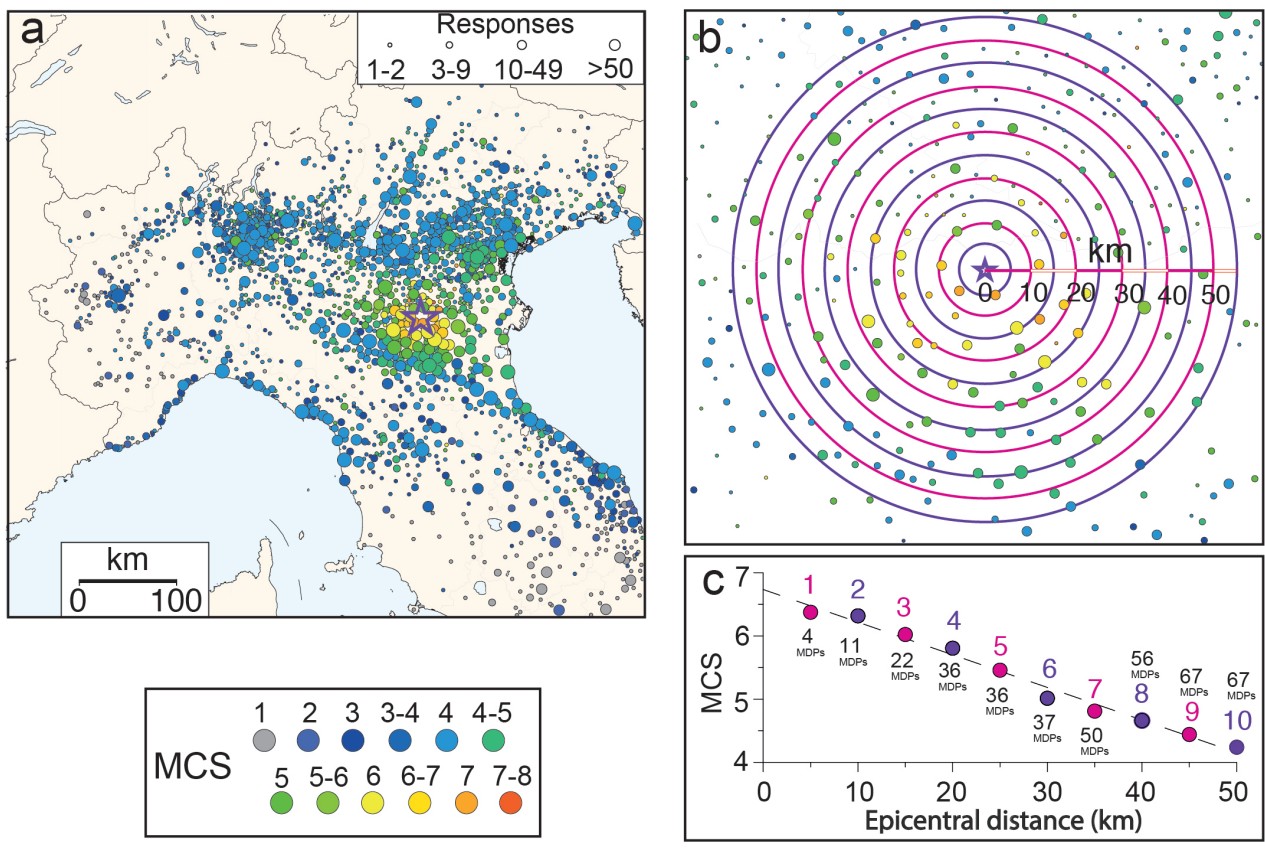

**Figure 2.** Workflow of the moving windows procedure. a) Macroseismic field of the 20 May 2012 earthquake (ID 13 in Table 1) from the HSIT database (available from: https://e.hsit.it/772691/index.html). The highest intensities are showed in the foreground; b) map showing the first 50 km from the instrumental epicenter and the ten ring-shaped search areas centered in the instrumental epicenter (shown by magenta and purple lines), each one shifted by 5 km with respect to the previous one; c) plot of the 10 intensity values obtained averaging the MDPs falling in each of the rings: #1 reports the average intensity calculated for the 0-10 km search area, #2 the average intensity for the 5-15 km search area, and so on.

**Table 1.** Full *learning set* used for this work. Events from 1 to 20 occurred in Northern Italy and were already used in the pilot work by Sbarra et al. (2019a). Latitude and longitude source parameters have been instrumentally obtained and derived from CPTI15 v2.0 catalogue (Rovida et al., 2019). Uncertainties about the epicentre location of longitude and latitude, reach a maximum value of ±2 and can be found at ISIDe catalogue (ISIDe Working Group, 2007). In the header "$M_w$" is the moment magnitude; "Lon" and "Lat" are the epicentral coordinates; "Depth unc" is the hypocentral depth uncertainty; "Std Err" is the standard error of the attenuation steepness; "$I_E$" is the expected intensity at the epicenter. The unit of the steepness and of the "Std Err" is (intensity/km).

| ID | Event date | Time (UTC) | $M_w$ | Source of $M_w$ | Lon | Lat | Hypocentral depth (km) | Depth unc (km) | Source of depth estimate | MDP (within 55 km) | MDP (all distances) | MDP source | Steepness | Std Err | $I_E$ |
|---|---|---|---|---|---|---|---|---|---|---|---|---|---|---|---|
| 1 | 9-nov-1983 | 16:29:52 | 5.0 | CSTI1.1 | 10.27 | 44.76 | 18.0 | – | CSTI1.1 | 231 | 850 | CFTI5Med | 0.040 | 0.0031 | 6.35 |
| 2 | 2-may-1987 | 20:43:53 | 4.7 | Italian CMT | 10.69 | 44.81 | 3.0 | ±3 | ISIDe | 175 | 802 | DBMI15 | 0.061 | 0.0037 | 6.52 |
| 3 | 10-may-2000 | 16:52:11 | 4.8 | Italian CMT | 11.93 | 44.24 | 13.1 | ±3 | ISIDe | 89 | 151 | Gasparini et al. (2011) | 0.068 | 0.0038 | 4.75 |
| 4 | 11-apr-2003 | 9:26:57 | 4.8 | Italian CMT | 8.87 | 44.76 | 8.2 | ±5 | ISIDe | 299 | 694 | Gasparini et al. (2011) | 0.048 | 0.0028 | 5.84 |
| 5 | 14-sep-2003 | 21:42:53 | 5.3 | Piccinini et al. (2006) | 11.38 | 44.26 | 20.1 | ±3 | Piccinini et al. (2006) | 100 | 133 | DBMI15 | 0.033 | 0.0021 | 5.74 |
| 6 | 26-mar-2008 | 9:19:30 | 4.0 | ISIDe | 9.81 | 44.34 | 72.2 | ±2 | ISIDe | 27 | 39 | HSIT | 0.017 | 0.0038 | 4.13 |
| 7 | 23-dec-2008 | 15:24:22 | 4.9 | ISIDe | 10.35 | 44.54 | 22.9 | ±1 | ISIDe | 85 | 670 | HSIT | 0.045 | 0.0066 | 5.67 |
| 8 | 5-apr-2009 | 20:20:53 | 4.5 | ISIDe | 11.91 | 44.23 | 24.5 | ±1 | ISIDe | 61 | 368 | HSIT | 0.021 | 0.0054 | 4.20 |
| 9 | 19-apr-2009 | 12:39:50 | 4.2 | ISIDe | 7.87 | 44.74 | 45.3 | ±2 | ISIDe | 200 | 384 | HSIT | 0.012 | 0.0020 | 3.57 |
| 10 | 13-oct-2010 | 22:43:14 | 4.0 | ISIDe | 12.38 | 44.21 | 26.5 | ±1 | ISIDe | 64 | 175 | HSIT | 0.023 | 0.0051 | 3.94 |
| 11 | 25-jan-2012 | 8:06:37 | 4.9 | ISIDe | 10.51 | 44.87 | 29.0 | ±1 | ISIDe | 160 | 1,354 | HSIT | 0.027 | 0.0031 | 4.80 |
| 12 | 27-jan-2012 | 14:53:13 | 4.9 | ISIDe | 10.01 | 44.52 | 72.4 | ±1 | ISIDe | 112 | 1,547 | HSIT | 0.007 | 0.0086 | 4.24 |
| 13 | 20-may-2012 | 2:03:52 | 5.9 | Govoni et al. (2014) | 11.26 | 44.90 | 6.3 | ±1 | Govoni et al. (2014) | 207 | 2,366 | HSIT | 0.052 | 0.0023 | 6.73 |
| 14 | 29-may-2012 | 7:00:03 | 5.7 | Govoni et al. (2014) | 11.07 | 44.84 | 8.1 | ±0 | Govoni et al. (2014) | 164 | 1,794 | HSIT | 0.062 | 0.0058 | 7.14 |
| 15 | 29-may-2012 | 10:55:57 | 5.3 | ISIDe | 10.98 | 44.87 | 8.7 | ±0 | Govoni et al. (2014) | 118 | 1,149 | HSIT | 0.059 | 0.0030 | 6.62 |
| 16 | 3-jun-2012 | 19:20:43 | 4.8 | Govoni et al. (2014) | 10.95 | 44.89 | 8.7 | ±0 | Govoni et al. (2014) | 167 | 1,512 | HSIT | 0.045 | 0.0035 | 5.68 |
| 17 | 6-jun-2012 | 4:08:31 | 4.0 | ISIDe | 12.32 | 44.40 | 31.1 | ±1 | ISIDe | 55 | 703 | HSIT | 0.023 | 0.0053 | 4.19 |
| 18 | 25-jan-2013 | 14:48:18 | 4.8 | ISIDe | 10.45 | 44.16 | 19.8 | ±1 | ISIDe | 142 | 1,129 | HSIT | 0.016 | 0.0028 | 4.42 |
| 19 | 18-nov-2018 | 12:48:46 | 4.0 | ISIDe | 12.49 | 44.05 | 36.8 | ±0 | ISIDe | 90 | 1,646 | HSIT | 0.005 | 0.0020 | 3.53 |
| 20 | 14-jan-2019 | 23:03:56 | 4.3 | ISIDe | 12.32 | 44.37 | 20.6 | ±1 | ISIDe | 60 | 1,748 | HSIT | 0.031 | 0.0051 | 4.47 |
| 21 | 5-may-1990 | 07:21:29 | 5.8 | Di Luccio et al. (2005) | 15.74 | 40.74 | 18.8 | ±0.3 | Di Luccio et al. (2005) | 139 | 1,375 | DBMI15 | 0.022 | 0.0023 | 6.38 |
| 22 | 26-may-1991 | 12:25:59 | 5.1 | Di Luccio et al. (2005) | 15.82 | 40.69 | 18.0 | ±0.3 | Di Luccio et al. (2005) | 137 | 597 | DBMI15 | 0.036 | 0.0022 | 6.39 |
| 23 | 15-oct-1996 | 09:55:59 | 5.4 | Selvaggi et al. (2001) | 10.68 | 44.80 | 15.0 | ±2 | Selvaggi et al. (2001) | 101 | 135 | DBMI15 | 0.040 | 0.0041 | 6.36 |
| 24 | 26-sep-1997 | 09:40:26 | 6.0 | Italian CMT | 12.85 | 43.01 | 8.0 | – | De Martini et al. (2003) | 135 | 869 | DBMI15 | 0.046 | 0.0030 | 7.71 |
| 25 | 26-mar-1998 | 16:26:17 | 5.2 | Olivieri and Ekström (1999) | 12.80 | 43.14 | 51.0 | ±4 | Olivieri and Ekström (1999) | 82 | 409 | DBMI15 | 0.014 | 0.0020 | 5.52 |
| 26 | 9-sep-1998 | 11:28:00 | 5.6 | Italian CMT | 15.95 | 40.06 | 29.2 | ±4 | Castello et al. (2006) | 115 | 689 | Gasparini et al. (2011) | 0.025 | 0.0024 | 5.96 |
| 27 | 31-oct-2002 | 10:32:59 | 5.8 | Vallée and Di Luccio (2005) | 14.89 | 41.72 | 16.0 | ±0.7 | Vallée and Di Luccio (2005) | 173 | 790 | Gasparini et al. (2011) | 0.048 | 0.0029 | 6.88 |
| 28 | 1-nov-2002 | 15:09:01 | 5.8 | Vallée and Di Luccio (2005) | 14.84 | 41.74 | 18.0 | ±0.1 | Vallée and Di Luccio (2005) | 165 | 638 | Gasparini et al. (2011) | 0.044 | 0.0022 | 6.35 |
| 29 | 21-oct-2006 | 07:04:10 | 4.2 | Italian CMT | 12.98 | 43.63 | 32.3 | ±5 | De Luca et al. (2009) | 66 | 100 | HSIT | 0.027 | 0.0036 | 4.90 |
| 30 | 6-apr-2009 | 01:32:40 | 6.1 | Chiaraluce et al. (2011) | 13.38 | 42.34 | 8.6 | ±1.5 | Chiaraluce et al. (2011) | 313 | 316 | Galli et al. (2009) | 0.043 | 0.0042 | 6.92 |
| 31 | 20-sep-2009 | 03:50:17 | 4.5 | ISIDe | 13.42 | 43.40 | 34.0 | ±1.5 | Cattaneo et al. (2017) | 128 | 427 | HSIT | 0.004 | 0.0017 | 4.06 |
| 32 | 15-oct-2010 | 05:21:19 | 4.3 | ISIDe | 16.63 | 38.88 | 37.2 | ±1 | ISIDe | 62 | 114 | HSIT | 0.027 | 0.0060 | 3.84 |
| 33 | 28-aug-2012 | 23:12:15 | 4.5 | ISIDe | 15.73 | 38.20 | 48.9 | ±1 | ISIDe | 91 | 318 | HSIT | 0.015 | 0.0038 | 4.21 |
| 34 | 25-oct-2012 | 23:05:24 | 5.2 | ISIDe | 16.02 | 39.88 | 9.7 | ±1 | ISIDe | 97 | 546 | HSIT | 0.035 | 0.0043 | 5.13 |
| 35 | 5-dec-2012 | 01:18:20 | 4.1 | ISIDe | 13.66 | 42.91 | 23.1 | ±0.7 | Cattaneo et al. (2017) | 119 | 285 | HSIT | 0.049 | 0.0041 | 4.44 |
| 36 | 16-feb-2013 | 21:16:09 | 4.8 | Frepoli et al. (2017) | 13.57 | 41.71 | 18.6 | ±0.9 | Frepoli et al. (2017) | 191 | 735 | HSIT | 0.025 | 0.0030 | 4.77 |
| 37 | 29-dec-2013 | 17:08:43 | 5.0 | ISIDe | 14.44 | 41.39 | 20.4 | ±1 | ISIDe | 219 | 879 | HSIT | 0.035 | 0.0054 | 5.07 |
| 38 | 24-aug-2016 | 01:36:32 | 6.0 | Michele et al. (2020) | 13.23 | 42.70 | 7.9 | ±0 | ISIDe | 142 | 143 | Tertulliani and Azzaro (2016) | 0.040 | 0.0100 | 6.99 |
| 39 | 30-oct-2016 | 06:40:17 | 6.5 | Michele et al. (2020) | 13.11 | 42.83 | 7.3 | ±0 | ISIDe | 236 | 241 | Tertulliani and Azzaro (2016) | 0.051 | 0.0021 | 8.06 |
| 40 | 18-jan-2017 | 10:14:09 | 5.5 | Michele et al. (2020) | 13.28 | 42.53 | 8.4 | ±0 | ISIDe | 66 | 67 | Tertulliani and Azzaro (2017) | 0.034 | 0.0075 | 7.26 |
| 41 | 16-aug-2018 | 18:19:04 | 5.1 | ISIDe | 14.87 | 41.87 | 19.6 | ±1 | ISIDe | 110 | 1,214 | HSIT | 0.041 | 0.0047 | 5.27 |
| 42 | 6-oct-2018 | 00:34:19 | 4.6 | ISIDe | 14.88 | 37.63 | 5.5 | ±1 | ISIDe | 71 | 253 | HSIT | 0.060 | 0.0100 | 5.11 |

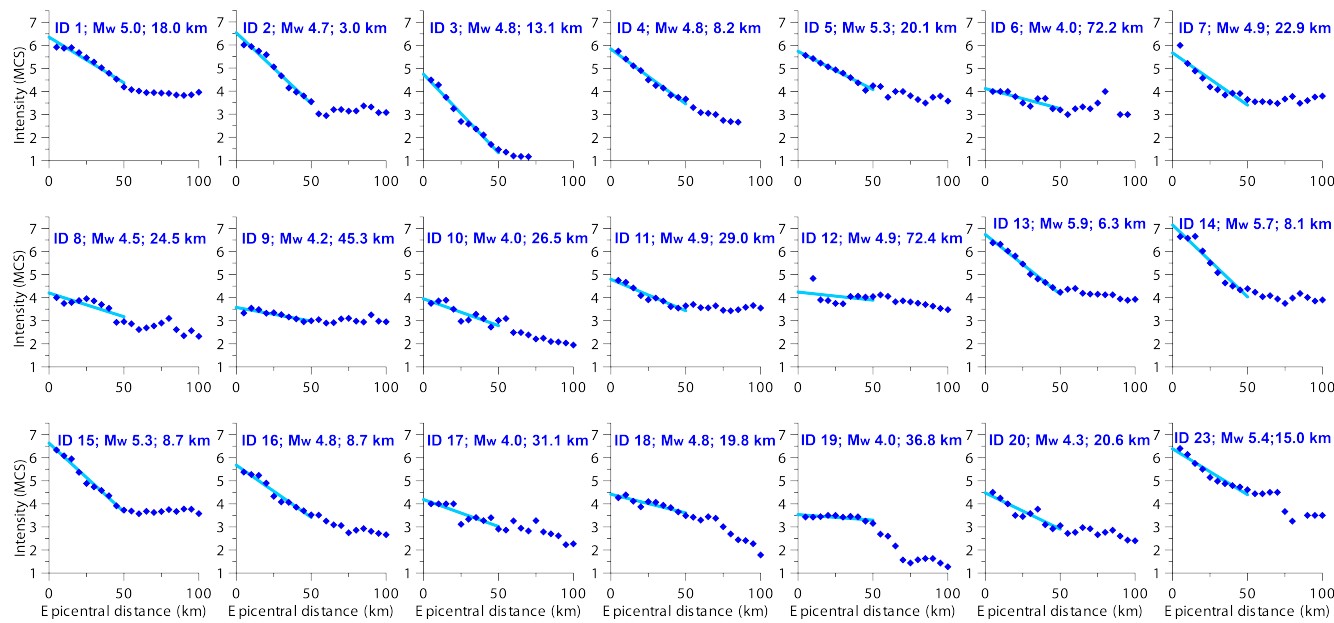

**Figure 3.** Attenuation curves obtained for the northern Italy earthquakes of the *learning set* (Table 1, from #1 to #20, plus #23; blue symbols in Figures 1, 5. Individual intensity data points were obtained by averaging the intensity values as shown in Figure 2. We obtained the linear fit for the first 50 km of each curve and calculated the resulting steepness.

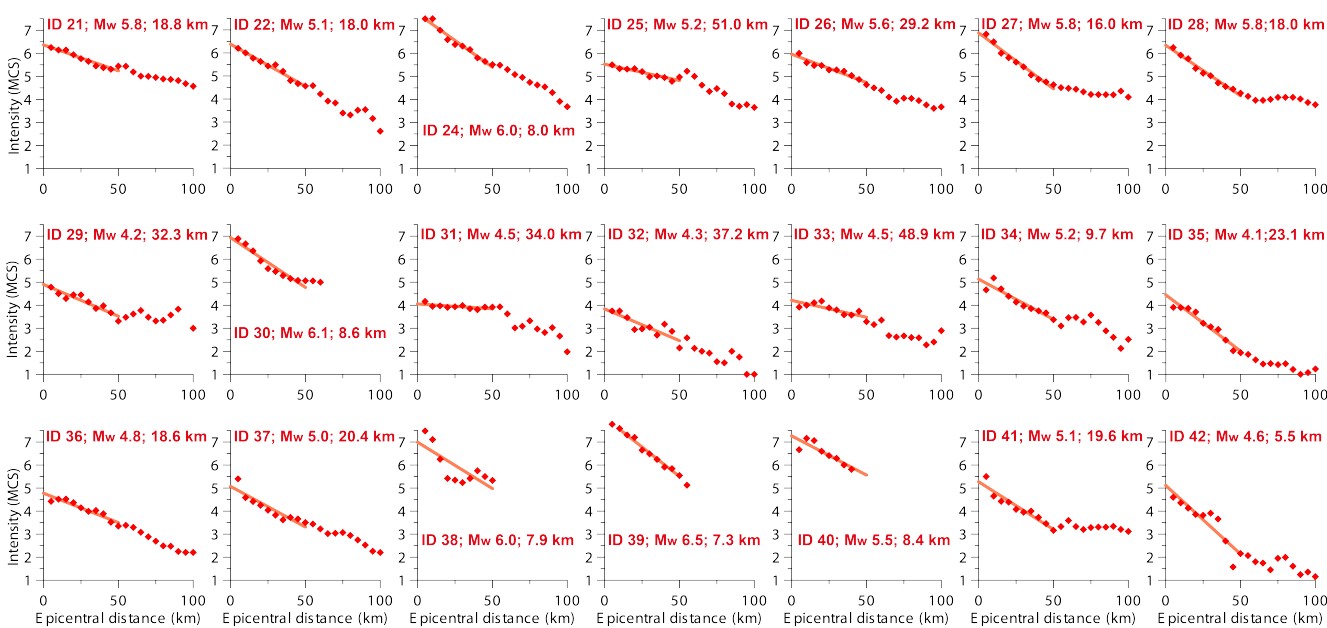

**Figure 4.** Attenuation curves obtained for the central and southern Italy earthquakes of the *learning set* (Table 1, from #21 to #42, except for #23; red symbols in Figures 1 and 5). Individual intensity data points were obtained by averaging the intensity values as shown in Figure 2. We obtained the linear fit for the first 50 km of each curve and calculated the resulting *steepness*.

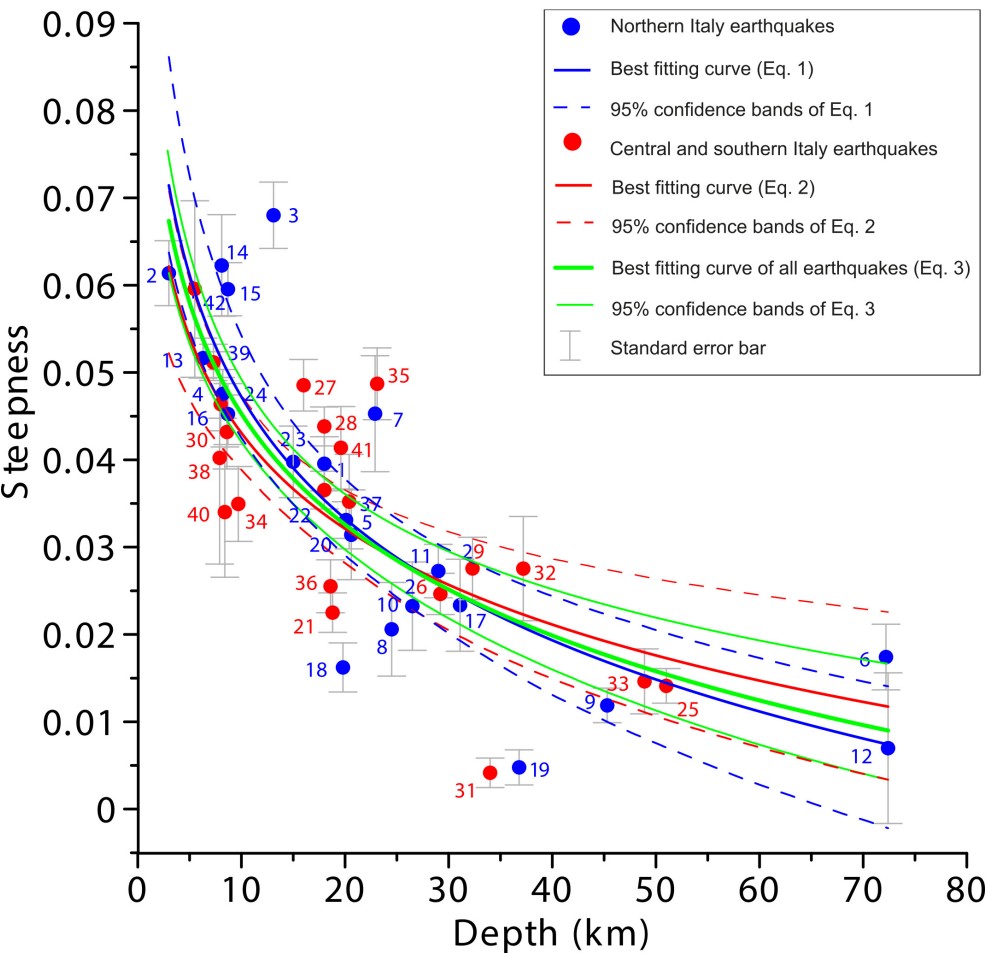

**Figure 5.** Depth versus attenuation steepness for the 42 earthquakes used as a *learning set*. Blue and red symbols indicate the northern Italy and the central and southern Italy datasets, respectively: the corresponding best fitting logarithmic functions are shown in blue (Eq. 1) and red (Eq. 2), respectively, along with their 95% confidence bands. The best fitting function obtained for the whole dataset (Eq. 3) is shown in green. Each earthquake is labelled with a unique identifier (see Table 1) and is plotted along with its standard error (shown by a vertical bar of ± standard error).

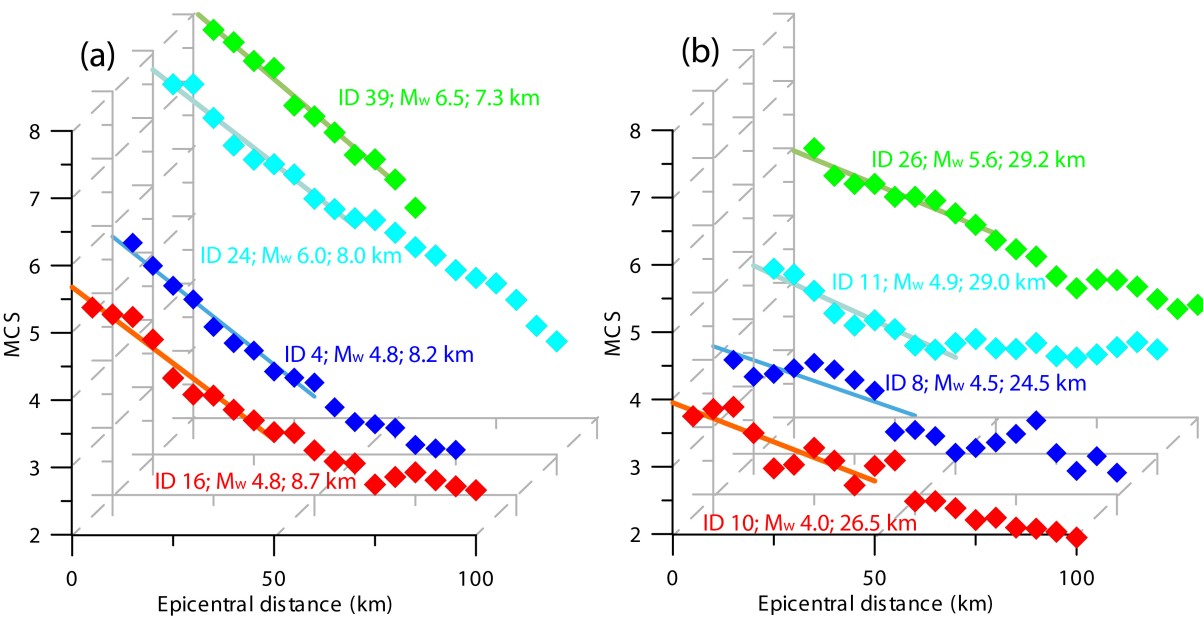

**Figure 6.** Attenuation curves obtained for two groups of earthquakes featuring a similar hypocentral depth but a different magnitude: (a) for the shallow events #4, 16, 24, and 39; (b) and for the deep events #8, 10, 11 and 26 (see Table 1 for further details). The steepness of the best-fitting line in the first 50 km is similar among the four events reported for each group, providing empirical evidence for the independence of inferred depth from magnitude in our methodology.

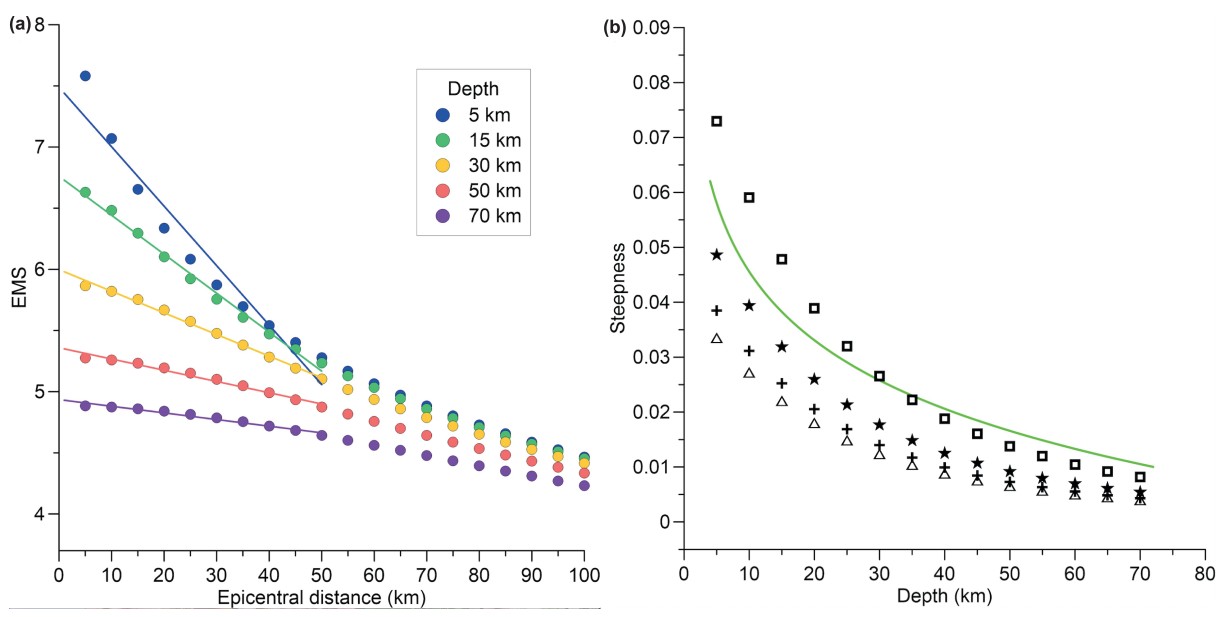

**Figure 7.** Attenuation curves and steepness simulated with different intensity or ground motion models for a $M_w$ 5.0 earthquake located at increasing depths. a) averaged intensity calculated using the IPE by Musson (2013) and the corresponding regression lines. b) attenuation *steepness* of intensities calculated using the IPE by Musson (2013) (stars), the IPE by Tosi et al. (2015) (crosses), and the GMPE by Cauzzi and Faccioli (2008); the PGAs predicted by this latter equation have been converted into MCS using the equations provided by Faenza and Michelini (2010) (squares) and by Masi et al. (2020) (triangles), respectively. All predictions are compared with the empirical Eq. 3, shown by the green line in panel b.

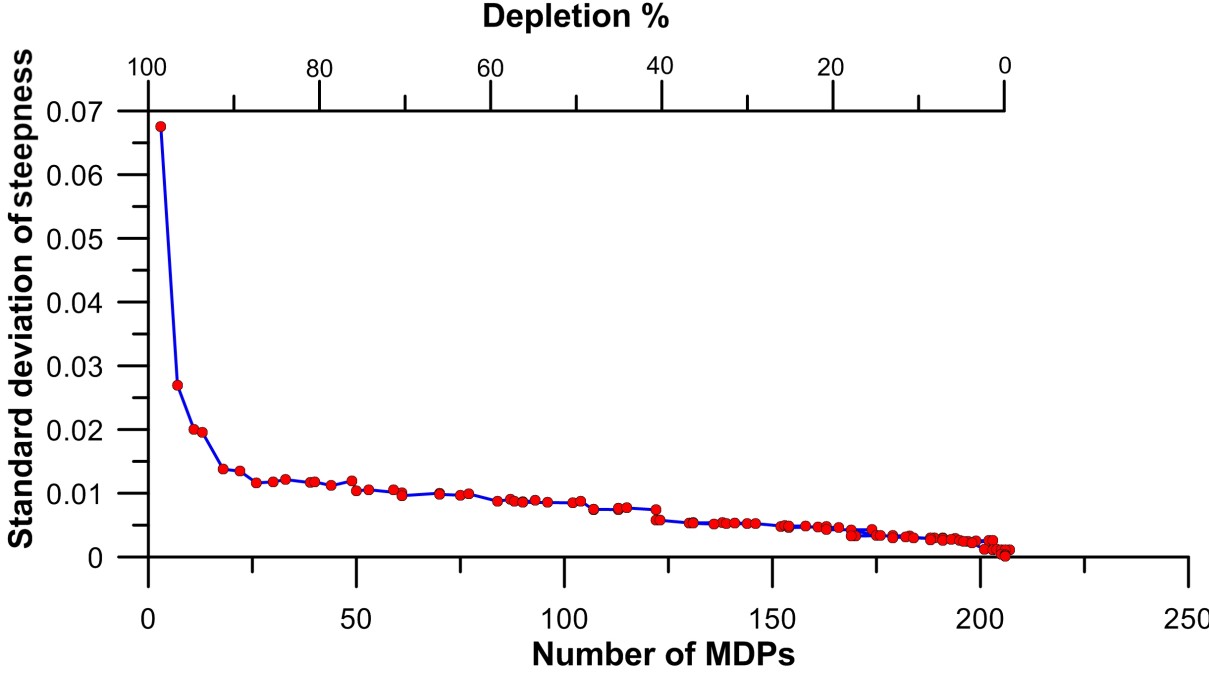

**Figure 8.** Application of the depletion test to the macroseismic field of the 20 May 2012 earthquake (Figures 1, 2; ID 13 in Table 1) taken from the HSIT database (available from: https://e.hsit.it/772691/index.html), for which there exist 207 MDPs falling within a radius of 55 km from the epicenter. The % of MDPs falling within each ring-shaped area (see Figure 1) was gradually depleted from 1% to 99%, and the steepness was calculated 1,000 times for all the different depleted datasets. So a total of 100,000 calculation was done. The Y-axis shows the variability of the calculated steepness, expressed through the standard deviation of the steepness obtained for the depleted datasets with the same number of MDPs.

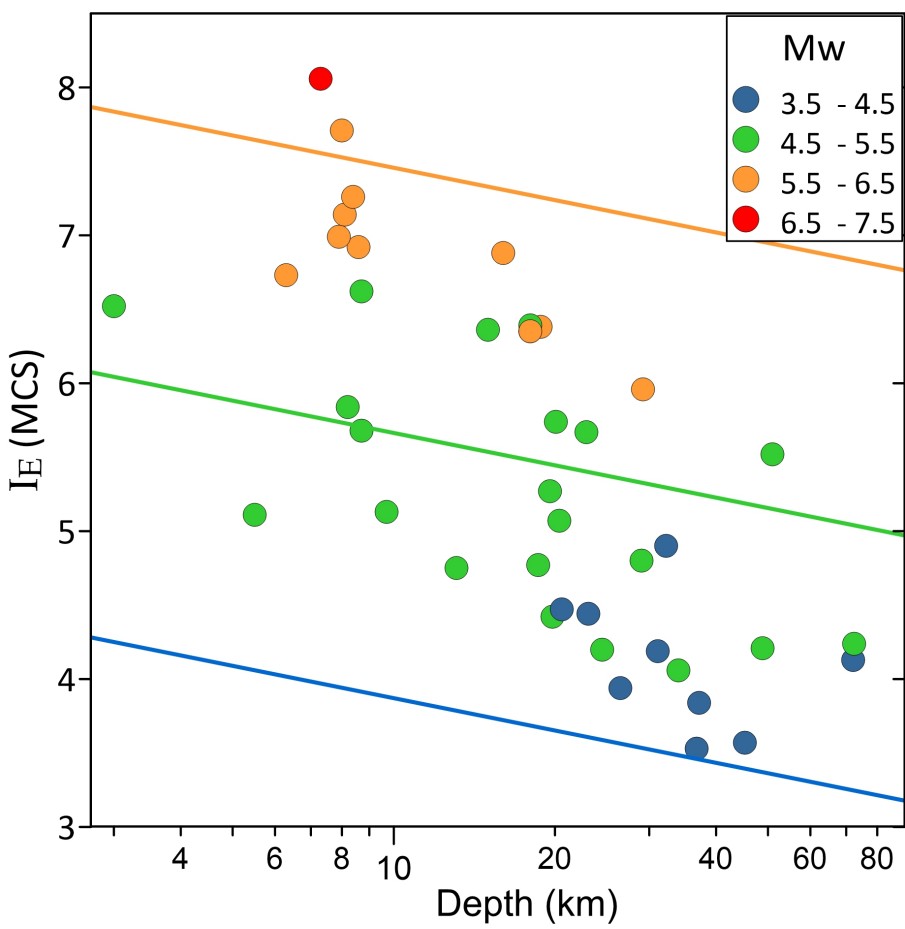

**Figure 9.** Magnitude as a function of the natural logarithm of depth, and expected intensity at the epicenter $I_E$ for all earthquakes of the *learning set* (colored dots). The multiple regression function (Eq. 4) is shown with colored lines of equal magnitude for Mw 4.0 (blue), Mw 5.0 (green), and Mw 6.0 (orange).

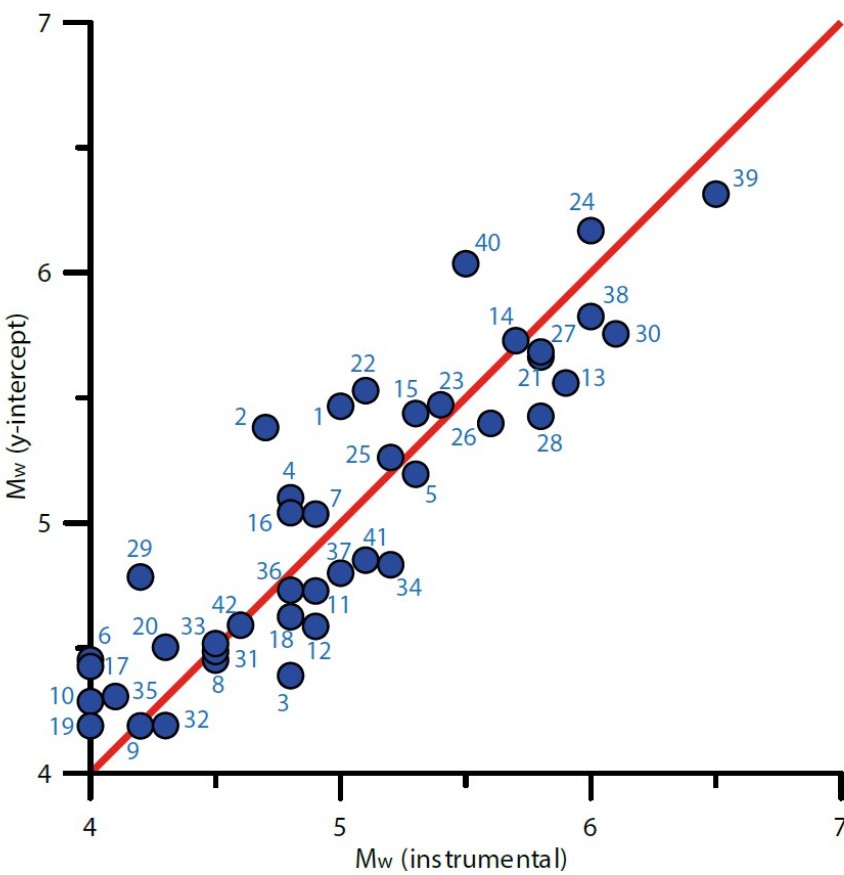

**Figure 10.** Correlation between the $M_w$ calculated with the *y-intercept* approach proposed in this work and the instrumental $M_w$ reported in Table 1 (Rovida et al., 2020) for all the events of the *learning set*. Earthquakes falling above or below the line exhibit a larger (up to +0.61 magnitude units) or smaller (up to -0.41 magnitude units) $M_w$, respectively.

**Table 2.** Comparison of $M_w$ estimates. The source of $M_w$ (Boxer code) is the CPTI15 catalogue, v2.0 (Rovida et al., 2019), with the only exception of the last three events, whose $M_w$ is from Rossi et al. (2019). The $M_w$ (*y-intercept*) is from this work.

| Event date | Time UTC | $M_w$ (instrumental) | $M_w$ (*y-intercept*-This work) | $M_w$ (Boxer code) | Source of instrumental $M_w$ |
|---|---|---|---|---|---|
| 9-Nov-1983 | 16:29:52 | 5.0 | 5.47 | 5.14 | CSTI1.1 |
| 2-May-1987 | 20:43:53 | 4.7 | 5.38 | 4.91 | Italian CMT |
| 26-May-1991 | 12:25:59 | 5.1 | 5.52 | 5.22 | Di Luccio et al., 2005 |
| 15-Oct-1996 | 09:55:59 | 5.4 | 5.47 | 5.19 | Selvaggi et al., 2001 |
| 26-Sep-1997 | 09:40:26 | 6.0 | 6.17 | 5.89 | Italian CMT |
| 10-May-2000 | 16:52:11 | 4.8 | 4.39 | 4.40 | Italian CMT |
| 1-Nov-2002 | 15:09:01 | 5.8 | 5.43 | 5.21 | Vallee and Di Luccio, 2005 |
| 31-Oct-2002 | 10:32:59 | 5.8 | 5.68 | 5.33 | Vallee and Di Luccio, 2005 |
| 14-Sep-2003 | 21:42:53 | 5.3 | 5.19 | 4.83 | Piccinini et al., 2006 |
| 6-Apr-2009 | 01:32:40 | 6.1 | 5.75 | 6.19 | Chiaraluce et al., 2011 |
| 20-May-2012 | 02:03:52 | 5.9 | 5.56 | 5.15 | Govoni et al., 2014 |
| 29-May-2012 | 07:00:03 | 5.7 | 5.73 | 5.43 | Govoni et al., 2014 |
| 24-Aug-2016 | 01:36:32 | 6.0 | 5.82 | 6.46 | Michele et al., 2020 |
| 30-Oct-2016 | 06:40:17 | 6.5 | 6.31 | 7.00 | Michele et al., 2020 |
| 18-Jan-2017 | 10:14:09 | 5.5 | 6.03 | 5.60 | Michele et al., 2020 |

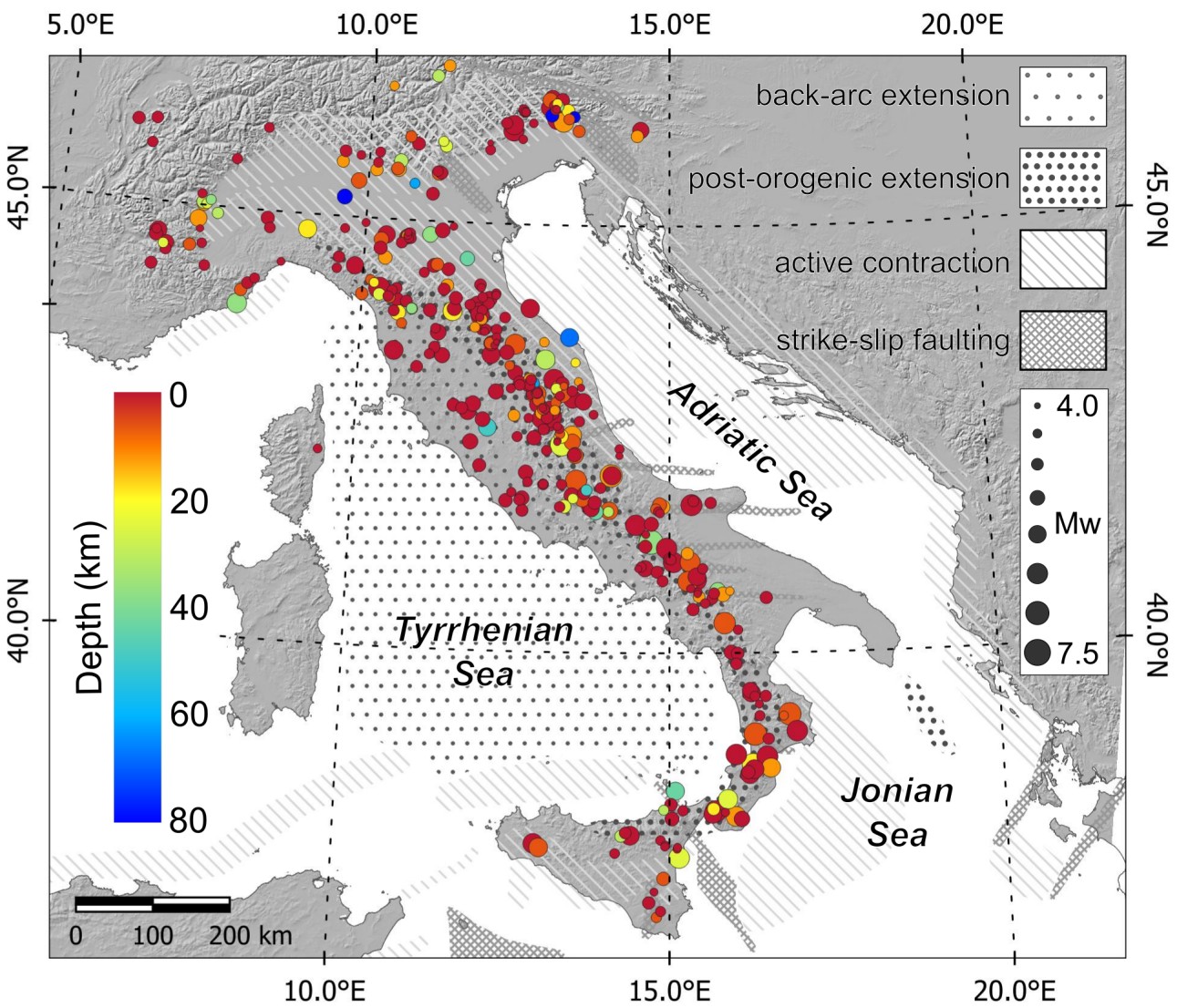

**Figure 11.** Estimated depth calculated using our approach (colour coded) for the 206 earthquakes of the *analyzed set*, shown with symbol size scaled with the magnitude calculated in this work.The areas with different patterns indicate active tectonic domains that exist in the Italian peninsula and surrounding areas (same as Figure 1); (from Vannoli et al., 2021).

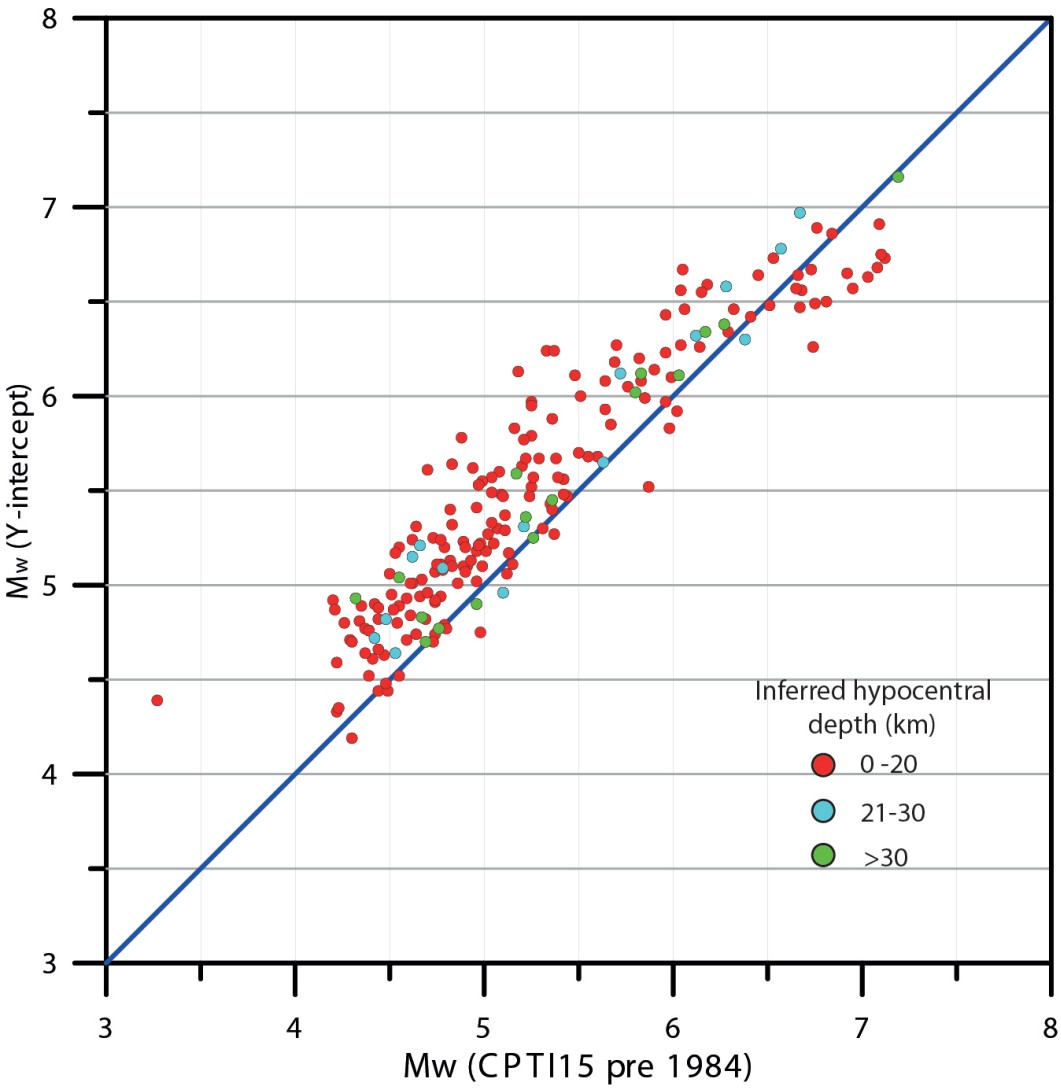

**Figure 12.** Correlation between the $M_w$ calculated with the *y-intercept* approach proposed in this work and the $M_w$ reported in the latest version of the pre-1984 CPTI15 catalogue (Rovida et al., 2021) for the 206 events of the *analyzed set* (Table S1). Earthquakes falling above or below the blue line exhibit a larger (up to +1.12 magnitude units) or smaller (up to -0.48 magnitude units) $M_w$, respectively. The global average is +0.25 magnitude units.