# Peer review of "Inferring the depth and magnitude of pre-instrumental earthquakes from intensity attenuation curves"

_Natural Hazards and Earth System Sciences, 2022_

## Author Response (AR1)

Dear NHESS Editor:

please find attached the revised version of our paper "Modern earthquakes as a key to understanding those of the past: the intensity attenuation curve speaks about earthquake depth and magnitude", by P. Sbarra, P. Burrato, V. De Rubeis, P. Tosi, G. Valensise, R. Vallone and P. Vannoli. The paper was substantially revised following the suggestions of Reviewers #1 and #2, whom we wish to thank for their dedication which definitely improved the manuscript.

In particular, we tried to clarify the trade-offs between depth, magnitude and attenuation properties of seismic waves, by discussing them more extensively. Moreover, we reorganized the text and added a new paragraph on the influence of cumulative macroseismic effects on the depth and magnitude estimation. We also added as additional material all 206 histograms and graphs of the attenuation curves for all earthquakes of the analyzed set.

We hope that we addressed all comments properly and that  this version of the manuscript may be considered for publication.

Thank you and best regards,

Paola Sbarra, on behalf of all the Authors

**Point by point response to reviewers' comments**

| Rev. #1 | |
|---|---|
| The authors of this manuscript present a procedure based on seismic intensity to determine first the focal depth of earthquakes in Italy and on a second step to determine the magnitude once the depth is assessed. The procedure is an extension for the whole Italy of that developed in a former work of the authors which was applied to earthquakes in Northern Italy. | First of all, we wish to thank this anonymous Reviewer #1 for his/her comments, which we found very pertinent and stimulating. We considered his/her detailed suggestions in the revised version of the manuscript, which has undoubtedly been improved in terms of clarity and readability. |
| **Main Comments** | |
| The manuscript seems hasty and the text is not precise and/or formal enough in many parts. Also, the manuscript has too many references to Sbarra et al (2019a) assuming the reader is familiar with it. Even if the present manuscript is an extension of the work of Sbarra et al (2019a), it should be self-explanatory by itself. Thus, the manuscript needs extending descriptions and/or giving enough details whenever necessary so that the reader is able to follow it. | This work extends the work by Sbarra et al. 2019a and includes a new experimental method to calculate the earthquake magnitude. We made the work self-explanatory, avoiding to make reference to our previous work too often, and adding the necessary details. |
| The structure of the manuscript is confusing for the reader; it would need some reorganization. I would suggest e.g.: Introduction | We reorganized the manuscript as follows: |

| | |
|---|---|
| Seismotectonic complexity and depth variability of Italian earthquakes

Methodology and data analysis (original section 3 adding a description of the method to estimate depth from the steepness)

      3.1 Data selection and analysis (original sections 3.1, 3.2, 3.3,3.4)

      3.2 Reliability and validation of the depth estimation method

      3.3 A two-step method for estimating magnitude based on intensity and depth (original sections 3.6 and 3.7)

      3.4 Reliability of the magnitude estimation method (original section 3.9)

      4. Application to the CPTI15/DBMI15 catalogues (original sections 3.8 and 3.10)

      5.  Conclusions | 1. Introduction

2. Seismotectonic complexity and depth variability of Italian earthquakes

3. Methodology and data analysis

   3.1 Data selection criteria

   3.2 Analysis of the learning set

   3.3 Independence of the method to infer the earthquake depth from magnitude

   3.4 Comparison with synthetic models

   3.5 Reliability and validation of the depth estimation method

   3.6 A two-step method for estimating magnitude based on intensity and depth

   3.7 Influence of macroseismic cumulative effects on the depth and magnitude estimation

   3.8 Dealing with larger magnitude earthquakes

   3.9 Reliability of the magnitude estimation method

4. Application to the CPTI15/DBMI15 catalogues

5. Conclusions

as suggested by Rev 1: |

| | We changed the header of section 3 in Methodology and data analysis.

Section 4. "Application to the CPTI15/DBMI15 catalogues" now contains original sections 3.8 and 3.10

As suggested by Rev. 2:

We changed the header of Section 3.3 into "Independence of the method to infer the earthquake depth from magnitude" and the header of 3.4 into "Comparison with synthetic models".

We also added a new section 3.7 "Influence of macroseismic cumulative effects on the depth and magnitude estimation". |
|---|---|
| Regarding the data and procedures more specific details and discussion is needed on the following: | |
| The text should incorporate a short description of the catalogues used and referred in the text. The 'analysed set' should be described in e.g., 3.1 | We briefly described all the catalogues used and cited in the text. |
| The manuscript needs a comment to qualitative nature of macroseismic intensity and the use of average intensities and rational intensity values instead of integer values. Are the averaged intensities normally distributed? I recommend a check. | We added a comment on the qualitative nature of macroseismic intensity in the introduction section.

In general, macroseismic intensity is far from being normally distributed, and we are well aware of the issues involved in treating the intensity as an integer or as a real (decimal) number. Macroseismic intensities are assigned after evaluating the effects of an earthquake at any given location. The resulting estimate is an integer, although the half-degree is often used even in direct field surveys in case of uncertainty between two contiguous degrees. This latter approach implies that intensity values must be processed as real numbers and that an uncertain assessment is either approximated to a half-integer, as proposed by Gasperini (2001), or simply discarded from the data set, as |

| | proposed by Albarello and D'Amico (2004). Nevertheless, assigning macroseismic intensities using web-based questionnaires entails greater precision, because it involves using decimal intensities rather than simply integer values (Wald et al., 2006). It has been demonstrated that this procedure leads to lesser scatter than if the calculated intensities were truncated to integers (e.g., Dengler and Dewey, 1998; Dewey et al., 2002). Thus, if on the one hand the macroseismic scales were designed as formed by a set of integer numbers, on the other hand, using decimal intensities allows for greater precision and lesser scatter. At any rate, both types of values must be dealt with within our work. |
|---|---|
| Both steepness and slope are used indistinguishably but formally the meaning is different. | We acknowledge the potential misunderstanding. Now we always refer to 'steepness'. |
| Because some of the relations are not supported with figures showing the data I suggest that the authors include some figures at least in the supplementary materials (see detailed comments). The location of the epicentre in the analysed events is never detailed, it seems both instrumental and macroseismic epicentres from the catalogues are used, but whether or not it would affect the results differently or the estimations of depth and magnitude is not addressed. | We added Figure S1 as requested. We have also added the 206 earthquake histograms (showing number of MDP's at distances less than 50 km) and 206 attenuation curve plots of the analyzed set to the .zip files of supplementary material S1 and S2 to complement Table S2. These graphs and the associated histograms allow the analyzed data to be examined in great details.. We understand that we were not clear enough in explaining the different use of the "preferred" epicentral location as supplied by the CPTI15 catalogue for our analysed set and consequently we have clarified this in the text. When the catalogue reports both an instrumental and a macroseismic epicentre, the choice of which of the two is "preferred" is made on a case-by-case basis by the catalogue compilers. Nevertheless, to minimise the ambiguities that may arise from these circumstances we analysed only pre-1984 events, for the vast majority of which the compilers selected the intensity-based magnitude and epicentre as |

| | |
|---|---|
| | preferred (Rovida et al., 2021). While this may influence the results, we preferred to stick to the choice made by the compilers, so as to warrant a more direct comparison with our results, also on the grounds that CPTI15 is an official 'reference database' for Italy. |
| In general, the formal statistical validation of the procedures and the formal estimation of uncertainties should be improved. | We added the estimation of the uncertainties of depth and magnitude for all earthquakes of the analyzed-set (table S1). Moreover we showed through multiple correlation between Mw, steepness, depth that steepness is only correlated with depth and not with Mw. |
| The text lacks discussions on critical and key issues such as: | |
| Possible distortion introduced by the instrumental epicentre/hypocentre in the computed distances and linear fits considering that the point where the fracture originates is not necessarily the point from which seismic wave energy radiates (Ground motion, macroseismic intensity). | A change in the location of the epicentre indeed affects the estimation of depth and magnitude, to an extent that depends on how big the change is. If the distance is on the order of 1-2 km, the resulting differences are negligible; in our case the maximum error for position in latitude and longitude is 2 km , while the instrumental error in depth exceeds 2 km only in a few cases (we added the uncertainties in Table 1 as given by the different sources).
This distance, however, may be larger for significantly larger events, but our dataset does not include earthquakes larger than M 6.5. |
| Choice of crossover distance of 50 km even for such seismotectonic complex region as specified in section 2 which would imply Moho depth variations and thus fluctuations on the distance at which reflected/refracted phases control the attenuation (see detailed comments). Some discussion and/or some estimation of the uncertainty introduced by the assumption (some trials to check the choice of 50 km crossover distance?) | In most cases, the trend of the attenuation curves for the 42 *learning set* earthquakes shows a substantial decrease in attenuation beyond an epicentral distance of about 50 km. This experimental result was confirmed by the work of Gasperini (2001) based on intensity data, and by Fah and Panza (1994), who used a numerical simulation of PGA attenuation, and was verified by Sbarra et al. (2019a), also based on macroseismic evidence. As shown by the attenuation curves given by the Intensity Prediction Equations (Fig. 7), 50 km is still a reasonable limit for a linear regression; we also added this observation in the text. |

| | |
|---|---|
| Learning set macroseismic data: How much does mixing HSIT and dedicated traditional studies affect the results of the learning set ? (see e.g., Hough, BSSA 103:2767-2781, 2013; Hough, BEE 12:135-155,2014) | The two datasets are complementary. The use of web-based data was fundamental to accomplishing our goals because these data were almost always the only available observations, especially for deeper earthquakes (>30 km) and smaller events. Furthermore, the use of macroseismic data obtained from direct surveys of earthquake damage was fundamental for the correct analysis of the attenuation curves, especially in the epicentral area. |
| | Intensity maps drawn for historical earthquakes exhibit more scattered damage patterns than those revealed by spatially-rich, web-based intensity data for similarly large events (Hough, 2013, 2014). This problem affects specifically those earthquakes whose effects are estimated through written sources. The same happens if only written sources (e.g., newspapers) are used to estimate intensities for recent earthquakes; they will inevitably end up being overestimated (Sbarra et al., 2010; Hough, 2014). The earthquakes included in our *learning set* are all relatively recent and the macroseismic field was estimated through a direct field survey, but, the problem delineated above does affect the *analysed set*. At any rate, assessing the quality of the macroseismic surveys available for historical earthquakes is beyond the scopes of this paper and will similarly affect the outcomes of any type of methodology designed to infer source parameters. |
| | We added a discussion about this topic. |
| Uncertainties in location (epicentre/depth) in learning set. Depth uncertainty is critical in the analysis. | We compiled our *learning set* discarding all the events whose hypocentral depth was fixed *a priori* and we used the hypocentral depth of the 42 events as taken from the official catalogue or from a reference paper dealing with the specific event. We |

| | added the uncertainties in Table 1 as given by the different sources. |
|---|---|
| Fitting of the slope-depth function. The curve is not constrained for depths above around 35 with only few data and apparently some outliers would need to be more extensively discussed. Also, while uncertainty in slopes is taken in account, depths are assumed not to be affected by uncertainty although the authors are well aware of it as mentioned in P13 L259-260 (see detailed comments P8 L170). | Yes, indeed. For distances larger than 35 km the uncertainty is inevitably greater. As a consequence, the confidence bands of EQ.3 in Figure 5 exhibits wider bounds, yet it still provides valuable information on depth estimation, albeit within a larger error range (We added the uncertainties in depth estimation in table S1). We added a discussion to clarify this issue. We neglected the uncertainty on instrumental depth because in most cases it is in the order of 1-2 km or less. See also the previous answer and see P8 L170 |
| Residual plots (Obs-Calc), not included in the manuscript, will greatly help to check for unbiased estimates of the empirical parameters. | We added the new figure S1 in the supplementary material; at any rate, the instrumental parameters and those estimated by our method (for *learning set* events) are listed in Table S2. |
| The comments and the suggestions in below are meant to improve the quality and readability of the manuscript and figures. | |
| **Detailed comments** | |
| **P3 L61-66**: Very unclear, this paragraph needs rewording and adding corresponding references to support the statements in the text. In its actual wording it is difficult to read and follow. Please explain better and refer to the literature. | We accepted this observation and added some references in the text. |
| **P3 L74**: The derived empirical equations relate the decay of intensity with distance (slope) with depth. They are not "Intensity-depth equations" as it reads. Please correct | Done |
| **P3 L78**: It reads "(Mw ≥ 6.75), because their causative fault cannot be assumed to be a point source…..". The assumption of a point source is valid or not depending on the distance from the source not only on the magnitude. Please elaborate and explain better. | Following the observation of the Reviewer #1 we modified the phrase and explained better why the assumption of the point source or not, depending on the magnitude of the specific event analyzed, can work in our approach. |
| **P4 L82-83**: Where throughout the manuscript "the role of crustal propagation properties versus the variability of depth" is evaluated as stated?. Please clarify. | We demonstrated this in the " Analysis of the learning set" section through EQ. 1 and EQ. 2 calculated only with the Northern and South-Central earthquakes, respectively. The coefficients of each function fall within the 95% confidence interval of the other function, suggesting that our method does not detect any statistically significant change in the macroseismic |

| | intensity attenuation properties between the two domains, at least over the first 50 km of epicentral distance. We clarified this result both in this section and in the conclusion. |
|---|---|
| **P6 L125-129**: It seems that quite a number of events in the learning set with depths > 30 km (e.g., 31, 19, 32.... ) do not fall in any of the four independent depth classes described (most of them are not in the Calabrian Arc) This is confusing. Please review. | Following the observation of the Reviewer #1, we changed the depth range of the third class of earthquakes because we actually do not know the exact maximum depth of this class, especially along the large lithospheric tears cutting through the Adriatic monocline and the Apennines. |
| **P6 L130-133**: I suggest to reword "… making difficult their interpretation just with the usual parameter determinations in the case of historical earthquakes for which only epicentre locations are given." | We thank the Reviewer #1 and accept this suggestion. |
| **P6 L137**: "well-located" is weak phrasing, how well? criteria, uncertainty, ….? | We rephrased as follows "the best available source for each event was chosen by performing an expert evaluation by analyzing all available literature sources." |
| **P6 L138**: "within 10 km-wide ring-shaped moving windows" Specify where the origin (0,0) of the rings is set. | This information is specified at the line 171. |
| **P6 L140-141**: I suggest to reword "… as described in Fah and Panza (1994) and Gasperini (2001) and empirically observed by Sbarra et al. (2019a) for earthquakes in Northern Italy". | Done. |
| **P6 L146-147**: "Notice that the size of our circular moving windows is now calculated from the earthquake epicenter rather than from the innermost MDP average, as proposed by Sbarra et al. (2019a)". Hasty? Size? Should it be "radius", or do you mean origin (0,0)? Please clarify "circular moving windows" or "circular rings moving windows"? | Following the suggestion of Reviewer #1 we rephrased the sentences to clarify. |

| | |
|---|---|
| earthquake epicentre" I assume instrumental?  It should read "instrumental earthquake epicentre"

"innermost" is innermost 10 km? specify Explain and clarify.

"MDP average" MDP average would be an intensity measure not the location of a point. Do you mean barycentre? Explain and/or clarify | |
| **P6-7 L147-148**: "This minor improvement makes the algorithm more uniform across the full earthquake magnitude range" why? Explain. | We added this explanation "avoiding translating all distances by a few kilometers from the epicenter". |
| **P7 L148**: "The new procedure" I would suggest new "approach". It is not really a new procedure | Done. |
| **P8 L169**: criterion #2 why? Explain | Deep earthquakes are more infrequent, and therefore the magnitude threshold of the latter is lower than the others to have a sufficient number of events to correlate steepness and depth (see Figure 5). |
| **P8 L170**: criterion #3 even if depth is instrumentally determined, it does not warranty the quality or small uncertainty in the calculated depth value. It seems a very weak criteria considering that it is one of the most critical parameters in the further analysis of the learning set and in the results. I would suggest to establish more strict criteria for the instrumental locations, especially in what regards depth determinations. | For the choice of depth of the 42 events comprising our *learning-set*, in addition to the criterion that the earthquake depth must not have been fixed *a priori*, for each individual event we performed an expert evaluation to choose the best available source. For each event of the *learning-set*, Table 1 reports the bibliographic source of its depth and magnitude (this piece of information was present also in the first version of our paper). Whenever a specific study about a given earthquake exists, we used the relocated depth (if available). We made sure that our *learning-set* contains only well-located instrumental earthquakes, i.e. events whose location uncertainties are small (especially for depth, whose uncertainty is indicated in Table 1). We have added this information in the text. |

| | |
|---|---|
| **P8 L171-172**: better "…..within a week time since the mainshock…." | We wrote "...within a week of the mainshoks". |
| **P8 L177**: "… six or more averaged points" I guess "in each of the rings"? specify. | We specified. |
| **P8 L179-180**: the reason for making an exception and include these two events #6 and #17 does not seem strong enough | The reason to include these two earthquakes with MDP<60 is that these are deep events that are crucial for characterizing lower crustal and subcrustal seismicity. |
| For section 3.1, it would be helpful an Appendix (e.g. as supplementary material) including some figures illustrating the geographical distribution of MDP's for each of the 42 selected events within 50 km distance from instrumental epicentre. | All 42 macroseismic fields are publicly available through the databases cited in the work:

CPTI15
https://doi.org/10.13127/DBMI/DBMI15.2
Locati et al., 2019

HSIT
https://doi.org/10.13127/HSIT
Tosi et al., 2007

CFTI5MED
https://doi.org/10.6092/ingv.it-cfti5
Guidoboni et al., 2018 |
| **P9 L194-195** Which latitude separates Northern Italy from the rest of Italy in the analysis?, and why this latitude? | Following the work of Mele et al, 1997, the latitude of 44° N is considered to divide two different macroseismic attenuation areas (see also Gasperini, 2001).
More or less at that latitude there is the geological and tectonic southern boundary of the Northern Apennines.
Besides, our previous paper used a dataset of earthquakes localized only in Northern Italy for the reasons explained in L231-234. |
| **P9 L196**: "… lithospheric structure and wave propagation properties are rather homogeneous ..." "rather" is weak phrasing, meaning? Include references in the literature. | Done. |

| | |
|---|---|
| **P9 L200**: "… slightly different …" This is weak phrasing. How much? Approximate range? Please quantify. | We quantified as follows: Mean=-0.00015; Max=0.007 |
| **P9 L207**: I suggest "… distance of about 50 km (see Figures 3 and 4)". | Done. |
| **P10 L223**: "… quite good …" Weak phrasing. Explain | We rephrased this sentence. |
| **P10 L225**: "… 0.058 ≤ S ≤ 0.012 …" formally should be 0.058≤ S ≤ 0.0097 or rounded, 0.058≤ S ≤ 0.010 | We corrected the typo. |
| **P11 L229-230**: "… In particular, the attenuation of earthquakes occurring in northern Italy, where the crust is …" Quite confusing sentence when looking at the plots in Figure 3, there is not a regular or homogeneous behaviour ( e.g. # 6, # 8, #19…….). Need to be more precise, rephrase maybe adding "typical", "frequent" | We rephrased. #6, #8, #19 are deep earthquakes, and in these earthquakes the bi-linearity between first section (<50 km) and second section (>50 km) of the attenuation curve is less evident because the first section is also shallowly sloping. However, in this sentence we are referring to the second section, we have clarified that. |
| **P11 L235**: "… implying that most likely it does not show in our analyses …" Weak phrasing, why do you assume this? To what small extend do you assume it? Surely negligible in the analysis? If different crust-mantle systems, some differences are to be expected even for crustal phases dominating closer distances and also on the transition distance form crustal phases to Moho reflected or refracted phases. | We rephrased this sentence. |
| **P11 L237-L238**: Add a new Figure with a dispersion plot showing the points Intensity-hypocentral distance and magnitude isolines. | We removed the IPE of old Eq. 4 as suggested by Reviewer #2, thus we did not add this figure. |
| **P11 L239-240**: "… This equation rests on the assumption that the macroseismic fields used to build it contain fairly well-distributed data, both in the near-field and in the far-field …" Weak phrasing. Does the assumption hold? "Fairly well" meaning? Are lower intensity data at long distances equally represented in the learning set? Are data complete for lower intensities in HSIT? Below which distance? These details should be specified and discuss and given to establish the range of hypocentral distances for which the relation can be considered valid. | Following the suggestion of Reviewer #2, we removed the new IPE from the revised version because it is not relevant to the results of the paper. |

| | |
|---|---|
| **P13 L249-250**: "The invariance of the attenuation slope with magnitude …" I suggest "The invariance of the attenuation slope with magnitude for the events shown …". Few examples as shown do not allow to extend this statement to all earthquakes (learning and analysed sets) and state "invariance" as it reads. Some more discussion (expected uncertainty at most introduced by this assumption) and more details are needed to support it which is key in the procedure following. | We showed the absence of correlation even numerically, explaining it by adding a sentence in the text. We empirically observe this behaviour for the *learning set* events, and we assume that it is the same for the *analysed set* by applying the principle of actualism. |
| **P13 L258-259**: "The endemic lack of interest for this parameter…….." I don´t think this statement is supported. I don´t find advisable at all this general judgement. Rephrase. | We rephrased this sentence as suggested. |
| **P14 L264**: "… magnitude-distance mixed term …" I suggest "… both magnitude and distance as one of the independent terms …" | Done. |
| **P14 266**: "… to turn the PGA …" I suggest "… to convert the PGA …" | Done. |
| **P14 L271**: I suggest "… using the IPE proposed by Musson (2005) for a magnitude M5.0 ... | Done. |
| **P14 L276-277**: "… It is worth noting that the differences caused by the use of the IPE in place of the GMPE are comparable to the differences caused by the use of two different conversion equations …" Very unclear sentence, please rephrase | We rephrased this sentence to clarify. |
| **P14 L288**: "… reliable estimate …" Weak phrasing. How "reliable" is defined? | We rephrased this sentence. |
| **P14 L291**: What's the criteria for reducing the number of MDP's? randomly?, azimuthally dependent? | Is randomly for each of the ten ring-shaped areas (of Figure 2b), we specified in the text that is not azimuthally dependent. |
| **P14 L292**: "… The regression of the attenuation …" should read better "… The regression linear fit of the attenuation …" | Done. |
| **P15 L294**: 0.01 is considered as the threshold for the standard deviation or for the standard error (in agreement with criterion #9 in P8)? Being in the order of magnitude of the steepness values, isn't it maybe too high for a threshold? | 0.01 in this line is, as stated in the text, the standard deviation of the distribution of 1000 steepness for each depletion step. In this case we used the standard deviation because we analyzed a distribution of data. We used the limit of 0.01 in |

| | |
|---|---|
| | accordance with criterion #9 because the units of standard error and standard deviation are the same. |
| **P15 L299**: Please explain how "reliable" is defined, in relation to uncertainty | We clarified in the text. |
| **P17 L325-326**: "… and the contour lines ..." I guess they are isolines of the function (magnitude isolines) not contour lines | Done. |
| **P17 326**: "… of the function that accounts for the geometrical spreading from the hypocenter to the epicentre ..." Explain better | We clarified in the text. |
| **P18-P19 L360-365**: Eliminate: "… which make it difficult to separate the individual contribution of a specific shock to the cumulative damage (Grünthal, 1998; Grimaz and Malisan, 2017; Graziani et al., 2019); a circumstance that would ultimately affect the attenuation slope and hence contaminate the inferred earthquake depth. This is a recurring problem in historical earthquake catalogues; a condition that is hard to overcome even for modern earthquakes, and even if a very rapid damage survey is carried out, because the first large shock inevitably causes an increase in the vulnerability whose effects on later shocks are virtually impossible to identify …" It does not directly relate and does not add to the explanation of the choice of the 7.1 magnitude earthquake. | We moved this part of text to a new paragraph as suggested by Reviewer #2. |
| **P19 L366**: The sentence should read: "… the RJB distance or using the moving window or the variable moving window …" | Done. |
| **P19 L367**: "… modest fluctuations …" Weak phrasing. Please quantify or give a threshold | We quantified in the following lines. |
| **P19 L367-368**: Which is the range of the errors arising from the uncertainties in the epicentral location? | We have modified the sentence, avoiding mention of epicentral location, because this comparison is not useful for our purposes. |
| **P19 L374**: The analysis of a single earthquake shouldn't support a general conclusion for the set of 21 earthquakes in the analysed set. | The reviewer is right that with only one example we should not statistically draw general conclusions, and we actually do not it, but many larger magnitude earthquakes of our dataset can not be used for the Joyner and Boore approach for many reasons, including the occurrence of multiple mainshocks in a sequence |

| | |
|---|---|
| | and the poor knowledge of the geometry and kinematics of their seismogenic source. |
| **P19 L375**: "… is not negligible …" Why?, range? Specify | We specified that the correction should be made for Mw >= 6.75 and the entity of the correction a few lines below. We added a supplementary zip file with the 206 attenuation graphs of the earthquakes of the analysed set. For the 21 earthquakes for which we assumed an extended source, we indicated the value of corrected and not correted $I_E$. |
| **P19 L386-387**: Eliminate the title of section 3.5. Not needed. | Done. |
| **P19 L387-388**: Add a figure in the supplementary materials including a histogram showing number of MDP's at distances less than 50 km for earthquakes in the analysed set (to complement table S1) | We added the 206 earthquake histograms of the analyzed set in a .zip S1 file in the supplementary material along with the 206 graphs of the attenuation curves (zip file S2) to complement Table S1. The additions of these graphs and histograms allow detailed control of the analyzed data. |
| **P19-21**: In section 3.8, a discussion of the results on depth and magnitude estimation should be added. | This paper was conceived as a methodological article; it aims at illustrating in detail a new approach for the calculation of magnitude and depth, using well-documented Italian earthquakes as a test case. For this reason, and also due to the length of our analysis, we did not develop the geological-structural interpretation of our results, but we plan to do it at a later time with a dedicated article as promised in lines 84-86. Also Reviewer #2 recommend to discuss these issues in a separate paper (https://doi.org/10.5194/nhess-2022-30-RC2). |
| **P22-23**: In section 3.10, a discussion of the results of the comparison should be added. | See previous answer. |
| **P22 L432-435**: The trend of the estimated y-intercept Mw is mostly above the CPTI15 Mw. CPTI15 Mw includes both instrumentally determined magnitudes together with macroseismic determined magnitudes. If according to Vanucci (2021), instrumental Mw might be overestimated this would explain the differences for part of the | The Boxer coefficients for calculating magnitude are calibrated using pre-1960 earthquakes, but in this case the difference between our estimates and the CPTI15 estimates would be even larger. The macroseismic moment magnitudes calculated using re-evaluated coefficients are, on average, lower than those |

| | |
|---|---|
| data. How would the overestimation be explained in the case of macroseismic determined magnitudes in CPTI15? | reported by the CPTI15 by 0.144.
The differences are due to the great diversity of methods. |
| **P24 L445**: Need to be more precise, e.g.: "… i.e. from the traditional macroseismic data in DBMI15 and from the new web based macroseismic data in HSIT dataset …" | Done. |
| **P24 L451-452**: According to L444 this conclusion holds for Italian earthquakes not for "any given earthquakes". Review. | Done. |
| **P24 L453**: Better "… Based on our learning set empirical observations …" | Done. |
| **P24**: The Conclusions should incorporate an important part devoted to the involved uncertainties if to be applied in seismic hazard studies. | We followed the suggestion of Reviewer #2 (see the comment "2.7. Length of the paper") and decided not to discuss seismotectonic and seismic hazard implications of our findings, leaving these topics to another paper that we are planning to submit. |
| **Figures and tables** ||
| Figure 2 caption: "… first 50 km from the epicentre … "… areas centered in the epicentre ..." Specify "instrumental" epicenter | Done. |
| Figures 3, 4:

Y_labels: should read Intensity (MCS)

Caption: No curves are shown in the plots, it is the data points together with the linear fit for distances less than 50 km. Reword the captions.

Besides the ID and depth in the inset in each individual graph include also the Mw and the number of MDP within 50 km (at least!, if not also the at all distances). I assume depths in insets correspond to instrumental depths, specify in reworded caption | Done.
We have included all the required information in the figures except the number of MDPs due to lack of space, however all this information is available in Table 1. |

| | |
|---|---|
| Figure 4: Depths in inset of each plot do not correspond with instrumental depths (I assume it is instrumental depths as in Figure3) in tables 1 and S2. Clarify and/or correct accordingly. | Yes is the instrumental depth. We have corrected some typos in Figure 4. |
| Figure 5: For clarity, I suggest to use smaller sizes of symbols and smaller fonts for text for the data points. Thicker line for the complete dataset curve, thinner for the two other curves. | Done. |
| Figure 6: As in Figures 3 and 4, include in insets Mw, Depth and number of MDPs. | We have included all the required information in the figures except the number of MDPs due to lack of space, however all this information is available in Table 1 |
| Figure 7a: Musson's IPE is developed for EMS intensities not MCS, review label in Y axis. | Done. |
| Figure 7b: Formally the green line should be Eq. 3 and not Eq. 6. Correct | Done. |
| Figure 9: The coloured lines should be labelled and/or explain it better in caption. "expected intensities" meaning? | Done. |
| In tables, include the meaning of acronyms in all table captions (e.g. ISIDE, HSIT, DBMI15, CPTI15……..etc) even if this is described in text. | These are not acronyms, but are the names of the databases. |
| **Rev. #2** ||
| **1. General comments** ||
| The paper presents an approach to derive the hypocentral depth of historical earthquakes from macroseismic intensity observations. The method is calibrated using the instrumentally determined depth of recent earthquakes. Subsequently, in a second step, the moment magnitude of the historical earthquakes is derived, again by calibration, using the instrumental magnitude | We thank the anonymous Reviewer #2 for his/her interest in our work and for all comments and suggestions. We will follow his/her comments to make our text more understandable even for a non-specialist, and clarify some methodological steps that we did not describe in detail in the first version of the manuscript. Below we respond to the Reviewer's main comments. |

of recent earthquakes. Whether this step-by-step inversion of depth and magnitude or a joint inversion is superior, in general or in individual cases, remains to be discussed further.

The paper can be seen as a continuation of an earlier publication by the authors (Sbarra et al., 2019a with application to Northern Italy), in as much as the method is now applied for the entire Italian peninsula.

Magnitude and depth of historical earthquakes are relevant to assess earthquake hazard in Italy and elsewhere. The paper contains a substantial contribution to the evaluation of historical earthquakes and in turn to the assessment of earthquake hazard in Italy. Upon revision, this work will be suitable for publication in the Natural Hazards and Earth System Sciences (NHESS) journal.

My review is intended to contribute to an improvement of the paper. My review is mainly related to the readability, comprehensibility for a non-specialist, consistency of content, and relevance for publication in NHESS. I could not check all details, hence comments and suggestions in the review are not meant to be exhaustive. My review does not include verification, plausibility calculations or proofreading. I have not checked or validated the content of the equations, figures, tables, and references. -- In my review, I refer to the numbers of text line(s), figures, and tables of the manuscript. Double quotation marks ("...") denote text as quoted from the manuscript. Topics/issues referenced in a previous review by referee #1 (see RC1: 'Comment on nhess-2022-30', Anonymous Referee #1, 17 Mar 2022, https://doi.org/10.5194/nhess-2022-30-RC1) are generally not repeated or co-commented here.

| | |
|---|---|
| On the whole, I recommend a quality check and major revision of the paper by the authors before publication in NHESS. | |

**2. Specific comments**

| | |
|---|---|
| **2.1**. Title

In my personal opinion, the title of the paper is too general in the first part and too vague in the second. I would propose a title that reflects the content of the paper in a simple and specific way, similar to that of the pilot study Sbarra et al., 2019a, for example 'Inferring the depth and the magnitude of pre-instrumental earthquakes from macroseismic intensity data for Italy', or the like. | We discussed if changing or not the title of the paper, but we preferred to emphasize that our procedure is based on the analysis of earthquake intensity attenuation curves and that our purpose is to apply the principle of actualism |
| **2.2**. Pilot study 2019a

Apart from some minor updates in methodology, the paper is essentially an application of the Sbarra et al., 2019a method ("pilot work") to the whole territory of Italy. This should not be overlooked, and could be stated more explicitly in the abstract. However, this does not detract from the importance of the paper. The present work warrants publication in a unique form, not just as a follow-up. The paper should be self-explanatory (see also comment of referee #1), hence there should be no need to repeatedly refer the reader to the pilot study. It would be interesting to know a bit more about the differences in results compared to the pilot study and a statement as to which results are now regarded to be relevant for the N-Italy region. | We made sure that the work is self-explanatory, avoiding reference to our previous work too often. |

| | |
|---|---|
| **2.3**. Uncertainties

I would recommend some more discussion on uncertainties, for example: As the method is based on macroseismic intensity data, a statement about assessment and uncertainties of intensity in Italy would be helpful for the general reader. To what extent errors in intensity are transferred into those of inferred depth and magnitude of the analysed set? By contrast, what can be said about uncertainties of the instrumental data used in the learning set? In respect to error statistics, I suggest to explain somewhat more what has been calculated; further to use the standard terminology or, if in doubt, provide a formula (confidence interval, standard deviation or standard error, error bar, root mean square deviation, etc.). | As suggested, we have improved the discussion, particularly on the uncertainty associated with the input parameters of the method (macroseismic intensity and instrumental data). We have also included a statement on the complexity of calculating macroseismic intensities.
The errors on depth and magnitude caused by the error of intensity estimation, although of interest, are difficult to estimate for historical earthquakes. So we did not cover this topic which was beyond the scope of this paper.
Moreover we now added to the article the estimation of reliability of depth and magnitude by estimating the depths corresponding to the EQ 3 confidence bands for each calculated steepness (see also Figure 5) and on the base of those values we also calculated the uncertainties of the magnitude (Table S1).
We explained the statistics of uncertainties in better detail and we corrected statistical terminology where not appropriate.
We added in Table 1 the location uncertainties of depth. |
| **2.4**. Trade-offs

Possible trade-offs between depth and/or magnitude on one hand and seismic wave attenuation properties on the other hand are not clearly resolved in the paper, to my opinion. | Concerning the trade-offs between depth, magnitude and seismic wave attenuation properties, we have shown that the steepness of the attenuation curve of the earthquakes of the *learning set*, does not vary significantly between northern and central-southern Italy, at least over the first 50 km of epicentral distance (as shown in Figure 5). This result implies that our new function holds for the whole Italian territory, despite the well-known complexity of Italian geodynamics and the consequent geological heterogeneity. Moreover, we show that our method is independent of magnitude, meaning that the slope of the attenuation curve calculated within 50 km from the epicentre is affected only by earthquake depth, not by earthquake size. We maintain that the trade-offs among depth, magnitude and seismic wave attenuation properties are fully addressed in the manuscript. Nevertheless, we clarified this critical issue by discussing it more thoroughly in the text. |

| | Reviewer #2 maintains that a methodology based on joint inversion is more appropriate than a step-by-step methodology, precisely because of the known trade-off between magnitude and depth. For this reason, we exposed our results more clearly, based on experimental data, so as to make the readers fully understand the potential of our method, which may be used to estimate the depth of an earthquake from its macroseismic field without the need to know in advance its magnitude. |
|---|---|
| **2.5**. Large magnitudes

The authors recognise that the "point source approach" is limited and propose a correction for "larger magnitude earthquakes" (Mw 6.75 and above, Section 3.7 of the paper). My suggestion would be to consider treating earthquakes larger than a certain threshold in a case-by-case fashion (arguments see below). In any case, it would be helpful to present the min-max limits of intensity, magnitude, distance, depth, etc. for which the method is applicable, preferably in a small table. | We preferred an automatic approach even for large-magnitude events because we believe that the case-by-case approach is subjective and not repeatable in all cases. Applicability limits are given for the learning set in the "Data Selection Criteria" section. The maximum magnitude used is 6.5, as there are no higher magnitudes in the examined time period. For higher magnitudes we arranged to use a modified method because the point source approach is limited. We did not mention intensity limits because in the selection we have a minimum of magnitude 4.0, so the expected epicentral intensity will never be too low (about VI MCS); in Figures 3 and 4 this information can be derived. For the analyzed set, the same criteria apply, except for the number of MDPs, which decreases from 60 to 30. |
| **2.6**. Cumulative effects

The authors are aware that for earthquakes occurring closely in time and space (multiple events, strong aftershocks, etc.; see e.g. Graziani et al., 2019 and several others) there is a severe problem to assess macroseismic intensities for separate events individually, particularly in cases of historical earthquakes, and particularly for the larger ones. Macroseismic data may then reflect accumulated | We are aware of the role of cumulative effects for earthquakes occurring close in time and space. In fact, we already discussed this issue in the article. This question is crucial, especially for historical earthquakes for which intensities are derived through indirect sources (primarily written texts), leading to an even higher risk of confusing the effects of different events. As recommended, we prepared a separate Section named "Influence of macroseismic cumulative effects on the depth and |

| | |
|---|---|
| effects. The method to infer depth and magnitude may fail completely in such cases. As for the results of the paper, I recommend that this problem be discussed in summary in a separate section rather than in individual passages. | magnitude estimation" to this issue in the revised version of the manuscript. |
| **2.7**. Length of the paper

Notwithstanding the need to explain the method in general and some technical procedure in detail, the length of the paper could be shortened somewhat without loss of significance. Some repetition in the text could be avoided, and some less relevant or obvious details could be omitted, or moved to supplements (see my comments line by line below). A more stringent structuring of the manuscript would be helpful to make it easier to follow the 'red thread'  (see also comment of referee #1) and to avoid addressing the same topic repeatedly in different parts of the paper (e.g. data selection criteria, compilation procedures, etc.).  Furthermore, the consequences for the Italian earthquake catalogue, for seismic hazard and for seismotectonic implications in Italy resulting from this work are, in my opinion, beyond the scope of this paper. I recommend that these issues be discussed in a separate paper, since the target audience and objective of such discussions are different (and as promised in lines 84-86). -- On the other hand, some further explanations are needed in the paper (see details line by line below). Generally speaking, the figure and table captions could also be more detailed (as I think figures and tables should be understandable to some extent without reading the whole paper). | We removed unnecessary details. We conceived this paper as a methodological article. It aims to illustrate a new approach for calculating magnitude and depth, using well-documented Italian earthquakes as a test case. For this reason, and for the length of our analysis, we did not develop the geological-structural interpretation of our results, but we plan to do it in a dedicated future article as suggested by Reviewer #2. |
| **2.8**.  Wording, terminology, formal issues

Some more precision and uniqueness in wording would be helpful. Terms should generally be used in an invariable way (for example: 'hypocentral depth' (in short: 'depth'), 'moment magnitude' (in short: 'magnitude'), 'macroseismic intensity' (in short: 'intensity'), 'attenuation slope' (in short: 'slope'), etc.) to avoid possible | We have been more careful in using the same notation for the same parameter and the same concept to avoid possible confusion for non-specialists. We also tried to dispel any ambivalence in the case of terms for which a different name is required. |

confusions. Using variable terminology frequently for the same parameter, for what reason soever, may cause confusion (for example 'hypocentral depth' alternatively denoted as 'earthquake depth', 'focal depth', 'source depth', etc.). In particular, as steepness and slope denote the same parameter, either the term 'steepness' or the term 'slope' should be used, not both alternatively. I suggest to always use the same notation for the same parameter, if there is no reason not to. Also, the alternating way of denoting magnitude and depth results of the analysed set as being 'inferred', 'expected', 'estimated', etc. can occasionally cause confusion. Any possible confusion between analysed set versus learning set parameters as well as between macroseismic versus instrumental parameters should be avoided (see below).

On the contrary, whenever there is ambivalence a different designation is needed. The term 'attenuation', for example, is used in the paper mainly in the sense of decay of macroseismic intensity with epicentral distance but at some point also in the sense of seismic wave attenuation, an ambivalence that may cause confusion. In particular also, it would be helpful to always state whether parameters or values either have been derived from seismogram measurements ('instrumental') or from intensities ('macroseismic'); this concerns for example magnitude, depth, epicenter location, etc. A clear definition of the term 'pre-instrumental' and the term 'historical' is needed for understanding the paper.

For the general reader, all abbreviations, acronyms etc. should be explained, and referenced if necessary, the first time they appear in the text (for instance MCS, EMS; Mw, CE; as well as the data sources INGV, DBMI15, CPTI15, CFTI5Med, CSTI1.1, Italian CMT, ISIDe, DISS, IPSI, etc., a small table would be helpful for those); the same holds for terms that are not well known in general ('Rosetta stone', 'apparent magnitude', etc.). For figures, tables,

| | |
|---|---|
| and equations all physical quantities, parameters, and numerical values, as well as their respective errors, should be specified together with their symbols (if any) and physical units (if any), for example: 'depth D in km', 'slope S in km-1', etc., even though it has been done elsewhere already. For equations (formulas), the valid range of application should be specified. | |
| **3. Detailed comments** | |
| **Line 9-12**: How is this sentence to be understood? I presume it should be referred to the learning set ('… we observe for the learning set …'). Are all three observations, (1), (2), and (3), observed "rather unexpectedly"? For earthquakes beyond a certain size, observation (1) cannot be expected a priori, most obviously so for earthquakes that are both large and shallow (see the "larger magnitude earthquakes" in Section 3.7 and respective comments). On the other hand, I think that observation (3) could have been expected to some extent. | We rephrased the sentence. |
| **Line 15**: ... 'by elastic and anelastic attenuation', I suppose. | The text has been corrected as suggested. |
| **Line 21-22**: "... macroseismic intensity ... a rough proxy of a set of accelerometric records": I don't understand what this is trying to say (PGA?). | Yes, we mean that the individual MDP may be used as a proxy for PGA (or PGV). We added in the text the reference to the paper by Worden et al. (2012) that is used as reference by the USGS shakemaps, that uses the DYFI data points for calculating the seismic parameters at different locations (see also http://usgs.github.io/shakemap/manual4_0/ug_products.html). The same procedure is used by the Italian shakemaps calculated for historical earthquakes (http://shakemap.ingv.it/shake4/archive.html) for which they use only macroseismic data points. |
| **Line 25-26**: This statement is vague, at least, and possibly misunderstood with regard to the term "damaging earthquakes". To take an example: What is the "length of the instrumental record" | We included some references so as to make the phrase less vague. |

| | |
|---|---|
| in Italy and what is "the average recurrence interval" of an intensity 6 MCS earthquake in the whole of Italy, or, for an MDP of intensity 6 in Rome, for example? | We believe that, talking about these topics, we should also discuss the interval of completeness of the different catalogues (historical and instrumental), but this is beyond the scope of this work too.

In any case, to answer to the questions raised by Reviewer #2:
1. The INGV instrumental catalogues starts in 1985, when there was a well established national seismic network, but there exists the record of individual large earthquakes back to the beginning of the XX century.
2. The average recurrence interval of an intensity VI MCS earthquake in the whole Italy can be calculated from the historical catalogus. Using the CPTI15 catalogue we have 1566 earthquakes having I max higher than VI in the whole catalogue starting around 1000 CE (out of 4894 events), and 674 events having I max higher than VI after 1900 (out of 3129 events). So, on the average, there is a recurrence interval of 1.5 events per year considering the whole catalogue and of 5.6 per year since 1900.
3. To consider individual localities you can extract the seismic histories from the catalogues. For example Rome sustained 8 events with local intensity of VI or higher since the beginning of the catalogue (see also Galli and Molin, 2014; https://doi.org/10.1007/s10518-012-9409-0). |
| **Line 34, 63, 82, 219, etc**.: What is meant by "earthquake propagation", "propagation characteristics", "propagation properties", etc. ? Is it about seismic wave propagation? | The sentences refer to the propagation of seismic waves, it was specified in each sentence. |
| **Line 48 and 53**: At these points it is the trade-off (singular!) between magnitude and depth. | Yes we are discussing only about the trade-off between magnitude and depth. We corrected the text. |
| **Line 52**: What is the meaning of "apparent magnitude"? | We deleted the adjective "apparent" to make the phrase more clear and not give the idea that there exists another type of magnitude. |

| | We used the adjective "apparent" to say that for historical earthquakes, being the magnitude derived from the macroseismic intensities, and since these are controlled by the energy released and by the hypocentral depth (due to the attenuation of the seismic energy), the result is that two earthquakes of the same magnitude but different hypocentral depths, occurring in the same place, should have different macroseismic pattern and hence their magnitude apparently is different. |
|---|---|
| **Line 61-66, 82-84**: What can be said about mutual trade-offs (plural!) between magnitude, depth, and seismic wave propagation properties (in particular seismic wave attenuation)? What do the authors of this study and those of other studies think about the variability of relevant crustal properties in Italy and the possible influence on determining magnitude and depth of historical earthquakes? I suggest to clarify. | Based on our empirical observations we have shown that the steepness of the attenuation curve in the first 50 km from the epicenter does not vary much due to regional differences in seismic wave propagation properties (Figure 5), so at these distances there is only a trade-off between depth and magnitude. We have also shown that the steepness of the attenuation curve in the first 50 km from the epicenter is independent of magnitude and is solely a function of the source depth. We made this clear in the discussion and conclusion. |
| **Line 87-133**: Section 2 about seismotectonic complexity could be shortened (see my comment in 2.7. above). Figure 1 should be kept, however. | We think that Section 2 is needed in the frame of the manuscript to understand why there are seismogenic sources at different depth ranges and so why it is necessary to try to characterize the depth of historical earthquakes (with implications on their magnitude). For these reasons we decided to keep the chapter in its present length. |
| **Figure 1**: I suggest a figure caption text starting with: 'Location of the 42 earthquakes of the learning set ...', or the like. Mw is moment magnitude | Done. |
| **Line 138**: Does the averaging include weighting, for example based on the number of responses per MDP (as might possibly be surmised from Figure 2b)? | No there is no weighting, but a minimum number of MDPs is required as stated in section "Data selection criteria", selection criteria 6. |

| | |
|---|---|
| **Line 139**: Actually not "curves" but data points are calculated and shown. | We clarified the sentence. |
| **Line 139-142**: What is the explanation of the observed change in slope ("abrupt drop") of the intensity attenuation? If it has something to do with the Moho reflections between 50 and 100 km epicentral distance (Gasperini, 2001) shouldn't it be observed almost everywhere? | Fah and Panza (1994) interpreted this slope change, according to Suhadolc and Chiaruttini (1985), as the transition from direct Sg phase to one where several S-wave phases mostly reflected at the Moho gradually become part of the Lg wave train. We explained this topic in the section "Analysis of the learning set". The change in steepness is observed everywhere. What we are analyzing is the diversity of the steepness of the first section of the attenuation curve (0-50 km), which depends strictly on the depth of the earthquake. |
| **Line 141** and also **Line 243, 255, 278, 338, 453**, etc. and caption of **Figures 6 and 7**: What is meant by "experimental" at these points? Isn't it rather 'empirical' or 'observational'? | We used empirical as suggested by the Reviewer #2. |
| **Line 141-142**: I suggest to formulate once in detail, for example: '… we calculate the slope of the line that best fits the intensity average data points from 0 to 50 km epicentral distance (in short: "attenuation slope" or "slope") ... ', or the like. For graphical explanation it could be referred to Fig. 2c and subsequent ones. | We explained this concept at end of the section "Methodology and data analysis" |
| **Line 149-153**: I guess what is meant is '… each one is shifted …' and '... averaged MDP intensities ...'. I presume that the instrumental epicenter is used. I suggest to clarify. | We rephrased the sentence. |
| **Figure 2**: I suggest clarifying which legend belongs to which panel(s) in the figure. Does the "Responses" legend in Fig. 2a also apply exactly to Fig. 2b? In what order are the MDP data plotted in Fig. 2a? There is a chance that the red and blue colors in Fig. 2b (circles for the rings) and Fig. 2c (dots for the averaged intensities) are confused with the red and blue colors used in the | The "responses" legend in Figure 2a is the same of Figure 2b. We added this explanation "The highest intensities are shown in the foreground". We modified panels 2b and 2c by replacing red and blue colours with magenta and purple. We also added the number of responses in panel 2c. The average intensity can be read on the Y axis. |

| | |
|---|---|
| MCS intensity scale in Fig. 1a nearby. Moreover, the colors red and blue in adjacent Figures 1, 3, 4, and 5 again have a different meaning. Therefore, I suggest to consider changing colors for the circles in Fig. 2b and the dots in Fig. 2c, accordingly. I suggest adding: 'number of responses', 'intensity (MCS)', '… averaging the MDP intensities …', etc. | |
| **Line 166-181**:  It seems that quite a lot of criteria are necessary to form a successful learning set. Is the learning set, thus, a set of earthquakes that are particularly well suited to the method? How many earthquakes do not fit the 'linear fit of intensity attenuation up to 50 km' scheme? | We could have chosen a learning set by choosing earthquakes one by one, but we still wanted to establish criteria by which starting from the totality of events everybody could understand precisely which events were included and which were discarded, to make the process reproducible. However, the main criteria depend on the magnitude and the minimum number of MDPs, the choice of which was detailed in Figure 8 in section "Reliability and validation of the depth estimation method."  In addition, criteria 5 through 9 in the "data selection criteria" section are necessary for the calculated steepness to be reliable. Almost all events fit the bilinear attenuation pattern, with the change in slope around 50 km from the epicentre. Generally the deeper earthquakes show a higher standard error. |
| **Line 178**:  I suggest to specify the unit of the "attenuation slope" (km-1), and thus also that of its standard error (km-1). | We added the unit of attenuation slope "Steepness".
S = (intensity/km). |
| **Line 211** and following:  I presume that 'slope' and 'steepness' are effectively the same parameter S (as in the paper the absolute value is taken for both, see Figure 5, Table 1, etc.). Hence there is no sense to distinguish between these two terms (see my comment in 2.8. above). | We have been more careful to use the same notation for the same parameter and concept to avoid possible confusion for non-specialists. We have also tried to dispel any ambivalence in the case of terms for which a different name is required. |
| **Line 213** and following:  For Equation 1,  I suggest to denote that S is given in units of km-1 and D is in units of km. This holds for Eq. 2, 3,  and 6 as well. Bracketing the last two figures would be | We declare that S "steepness" is measured in "intensity/km"
We substituted the standard error value, for all the equations, with the 95% confidence intervals computed multiplying the standard error with the critical "t-student" value having fixed the |

| | |
|---|---|
| helpful for the appearance of the Eq. 1 to 4 (as it is done already in Eq. 5). How are confidences / errors calculated for Eq. 1 to 5? | significance level "alpha"= 0.05 (two-tailed test) and appropriate degrees of freedom. |
| **Line 217**: Is this referring to 'intensity attenuation' or to 'seismic wave attenuation'? | We specified "macroseismic intensity attenuation". |
| **Line 225-226**: I suggest to re-check numbers with Equation 3 and with green line in Figure 5. | We checked if the green line is correct. We changed the equations error because we decided to express the error using the confidence bands. |
| **Figure 4** and also **Fig. 3**: Comparing with Table 1 and Table S2, instrumental depth values (km) are expected to be given next to the event ID's in the insets (and should be mentioned in the figure caption accordingly). A spot check shows, however, that respective depth values in the tables are in a number of cases not the same as those in the figures (see also comment of referee #1). Apparently there is a need for a quality check and correction in the figures and/or in the tables, respectively. Intensity should be denoted as 'Intensity (MCS)'. | We have corrected some typos in figures 3 and 4. |
| **Line 229**: same as in line 217 | Intensity attenuation in this case. |
| **Line 229-231**: Are there no similar "plateaus" in the intensity attenuation data in central and southern Italy? What difference do the authors expect in the effect of Moho reflected waves between northern and central/southern Italy? | We have to clarify that this comment is about the second section of the attenuation curve, it. We have shown that in the first 50 km of the attenuation curve there is no influence of crustal attenuation properties. Instead, in general, the slope of the second section of the attenuation curve in Central and Southern Italy is higher than that in Northern Italy. We don't know if it depends on the Moho geometry, we are just reporting a hypothesis of the possible motivation. |
| **Line 233**: What is meant by "efficiency of the crust-upper mantle system"? | We clarified in the text the meaning of "efficiency of the crust-upper mantle system". |

| | |
|---|---|
| **Line 238-239**: For completeness of Equation 4, I suggest to add that Mw is moment magnitude, I is intensity (MCS), and to denote to which base 'log' is the logarithm (10?). | See next answer. |
| **Line 237-240**: How was Equation 4 derived? I would generally recommend that the derivation be documented when a "new intensity prediction equation IPE for Italy" is published here. A regression plot of the new IPE I(r, Mw) would be helpful as well (see comment of referee #1). Within what limits of r and Mw is the new IPE considered valid? Following the basic idea of the paper, shouldn't there essentially be two IPE's, one for the near and one for the far field? Finally, if I am not mistaken, the new IPE is not relevant for the results of the paper; nor is it used below, except for a casual comparison (in Section 3.5) with the IPE of Musson, 2005 (which applies to the UK, is not the most recent, and uses a different magnitude (ML) and intensity scale (EMS) anyway). | We agree that the new IPE is not relevant to the results of the paper and have removed it from the revised version. |
| **Figure 5**: How are the "95%-confidence intervals" determined? How is "standard error bar" determined, is "bar" twice the standard error? I suggest to use the label 'Slope S (km-1)'. | In the text we wrote more correctly 95%-confidence bands, they are calculated for each depth $x_k$ in this way $$\hat{y}_k \pm t_{cr} \cdot \sqrt{MSE\left(\frac{1}{n} + \frac{(x_k - \bar{X})^2}{\sum_{i=1}^{n}(x_i - \bar{X})^2}\right)}, \text{ where}$$ $$MSE = \frac{1}{n-2}\sum_{i=1}^{n}(y_i - \hat{y}_i)^2 \text{ and } t_{cr} \text{ is the critical t-student value}$$ We added the reference in the text. The bar is one standard error below and one standard error above the steepness value (in total twice the standard error). We added the unit of attenuation slope "Steepness". S = (intensity/km). |

| | |
|---|---|
| **Line 241**: I suggest to re-word the section header to: 'Independence of inferred depth from magnitude', to avoid misunderstandings. | We re-worded "Independence of the method to infer the earthquake depth from magnitude ". |
| **Figure 6**: same as for line 241 | Done. |
| **Line 249-253**: I suggest to maintain "invariance of attenuation slope with magnitude" to the earthquakes of the learning set in the first place. Whether this invariance holds for all earthquakes is an open question at this point (see also probable exceptions for the "larger magnitude earthquakes" discussed in Section 3.7). I also suggest to re-check and eventually re-word the statement "nearly all the methodologies developed in the past to calculate earthquake depth use magnitude as an essential input parameter", along with the references listed here in connection with this topic. | We specified that this assumption was verified for the earthquakes of the learning set, however, we assume it is the same for the analyzed set by applying the principle of actualism. We re-checked the statement. |
| **Line 254**: The header of Section 3.4 is inconclusive, in my opinion. It could be re-phrased to 'Comparison with synthetic models', or the like. | Done. |
| **Line 258-260**: To my opinion, there generally is no "endemic lack of interest" in hypocentral depth (see also comment of referee #1), but there frequently is lack of data to determine depth. | We rephrased this sentence as suggested. |
| **Line 268-269**: Which magnitude(s) have been used for the "hypothetical earthquake"? It should be stressed that Figure 7 is exemplifying the case of a "M 5.0 earthquake". Can the results of Section 3.4 be generalised for all magnitudes relevant in this paper? | Done. Yes can be generalised because the attenuation slope is independent by magnitude. |
| **Figure 7** and corresponding text in Section 3.4: I suggest to unify terms (see my comment above in 2.8.) What is the meaning of M (Mw, ML, or other, respectively)? I would suggest to refer the green line in Fig. 7b to Eq. 3, as it is done in Fig. 5. Data points in | M is Mw, we corrected in the text.
We refer the green line to Eq. 3.
We corrected Figure 7 as suggested. |

| | |
|---|---|
| Fig. 7b do not necessarily need colors, as they are coded by symbols already (colors in Fig. 7b must not be mistaken with that of Fig. 7a, anyway). | |
| **Line 262 and Figure 7**: The IPE of Musson (2005) uses ML whereas the IPE of Eq. 4 uses Mw. Why not take the updated IPE relation of Musson, R.M.W., 2013, Updated intensity attenuation for the UK. Nottingham, UK, British Geological Survey, 13pp. (OR/13/029), which is for Mw as well? Moreover, Musson (2005) and Musson (2013) are for EMS whereas IPE of Eq. 4 is for MCS, I presume. The agreement of the data from the IPE of Musson (2005) with that of the IPE of Eq. 4 is surprisingly good, though. | We substituted Musson (2005) with Musson (2013) and indicated EMS title in the y axis of Figure 7a. |
| **Line 276-278 and Figure 7**: The differences in slope caused by use of the two different conversion equations seem to be larger than any other difference from prediction models in Fig. 7b; I suggest to clarify and re-phrase eventually. Is there any idea why the empirical (observational) results for Italy (Eq. 3 and green line in Fig. 7b and Fig. 5) generally 'over-predict' the hypocentral depth for the deeper earthquakes compared to the (synthetic) prediction models shown here, or the other way round. Why, in particular, the slope S from Eq. 3 (Fig. 7b, green line) deviates considerably from the slope derived from the IPE Eq. 4 data even though the underlying data set is the same? To my opinion, the "trend" of the "curves" of the prediction models (the four dotted ones in Fig. 7b) is quite similar, for the empirical one (green line) it is not. | As suggested in Line 237-240 we removed the IPE of old Eq. 4. We added this sentence into the text. "The greater difference is observed for depths greater than 35 km probably because the empirical regression is less constrained at those depths. This is reflected in a wide confidence bands of EQ.3 (see Fig.5), due to fewer learning-set earthquakes at those depths" |
| **Line 285-286**: see my comment above (2.3.) | See 2.3. |
| **Line 290-301** and **Figure 8**: The "depletion test" procedure has not become clear enough, to my opinion (e.g. how many slope calculations in total?). I suggest to re-phrase and explain in more detail with reference to Figure 8. | The depletion test procedure was performed calculating the steepness 1,000 times for each percentage point of depletion from 1 to 99%, so a total of 100,000 calculations were done. |

| | |
|---|---|
| **Line 297-298**: I suggest, to clarify what is meant by "For deriving Eq. 3 we use an even more conservative selection of learning set data". Hence, is Eq. 3 derived from a sub-set of the learning set that is "even more" conservatively selected? The conditions for derivation of Eq. 3 should be presented elsewhere (in Section 3.1 and 3.2, I guess). | We have removed this sentence from here because it was an anticipation of what we state in section "Application to the CPTI15/DBMI15 catalogues" where we clarify this topic. The most conservative selection for the learning set concerns only the number of MPDs used. The conditions for the derivation of Eq. 3 have already been presented in section "Data selection criteria", criterion #6. |
| **Line 300-301**: What does "only a few MDPs" mean at this point? -- Taking 30 MDP's in 10 distance rings ("windows"), just for example, must there be an average of 3 MDP's "homogeneously distributed for each distance window"? | For the analyzed set, as mentioned, 30 MDPs is the minimum number. If the depletion procedure leads to a remaining number of MDPs less than 1 in a ring shaped area, 0 MDP is considered for that area (it doesn't make sense to consider in the calculation - let's say - 0.7 MDP). We added an explanation in the text. |
| **Line 302-307**: I recommend to clarify and re-structure the text concerning the various criteria applied to the learning set (42 events) and to the analysed set of earthquakes (206 events). It would be helpful to have the criteria for the analysed set clearly set out in listed form (similar to that of the learning set in Section 3.1); maybe in two versions, first: general selection criteria of the analysed set, second: stricter criteria (from averaging particular values of the analysed set) to distinguish a class of slope values that are significantly "more reliable" (see above). By the way, couldn't that be read from the standard error of the slope (Table S1) as well? | We reorganized the text. The earthquakes of the analyzed set must meet all criteria listed in Sect. "Data Selection criteria" except for #6, which we relaxed by reducing the minimum number of MDPs from 60 to 30, based on the conclusions drawn in Sect. "Reliability and validation of the depth estimation method", as stated in the section "Application to the CPTI15/DBMI15 catalogues". We also added in table S1 the uncertainties of depth and magnitude estimated with our method. |
| **Line 308-315**: The example given here is well suited to demonstrate that applying this method for earthquakes occurring closely in time and space poses a most severe problem to the method. See my comment above in 2.6. I suggest to add ID-numbers for reference with Table 1. | Done. We devoted a separate section to this issue in the revised version of the manuscript. |
| **Line 311**: There is a typo in 'http'. | Done. |

| | |
|---|---|
| **Line 317**: The statement "depth is independent of magnitude" needs a restrictive relation to the context. I suggest the wording: 'the hypocentral depth inferred for the analysed set is independent of magnitude up to a certain size', or the like. | We corrected the first part, we explained the second issue in the next section "Dealing with larger magnitude earthquakes". |
| **Line 318**: "affects the y-intercept" or 'is derived from the y-intercept', what is meant at this point? | We mean both, the estimation of the magnitude affects the value of the y-intercept (for estimating the expected intensity at the epicenter, $I_E$) and the magnitude for the *analyzed set* is derived using the value of the y-intercept itself. |
| **Line 320**: I suggest to complete: '... and decreases if depth increases for a constant magnitude'. | Done. |
| **Line 323-327**: I suggest to describe the regression analysis leading to Equation 5 in more detail (in comparison to the regression leading to the IPE, Eq. 4, see above). What is the valid range of application of Eq. 5? What is the difference (it seems to be large) between the relation Mw(r = D, I = IE) taken as a reversion of Eq. 4 at the epicenter on one hand and the relation Mw(D, IE) of Eq. 5 on the other? In other words, I suggest to explain for the reader why Eq. 5 has not just simply been derived from a reversion of Eq. 4 (in a way it apparently had been done in the pilot study Sbarra et al., 2019a). | Deriving magnitude using only well-studied earthquakes with their expected epicentral intensities gives us a better estimate of magnitude because it is based on higher intensities than that obtained by inverting an IPE . We specified this issue in the text. Moreover as suggested in Line 237-240 we removed the IPE of old Eq. 4. Therefore, we did not make comparisons between the 2 equations
Eq. 5 (new Eq. 4) can be applied for a depth interval 5 ≤ D ≤ 73 km and 3.5 ≤ IE ≤ 8.1 we specified in the text. |
| **Line 325-326**: What is meant by "the contour lines of the function that accounts for the geometrical spreading from the hypocenter to the epicenter"? Why just "geometrical spreading"? I suggest to clarify. | We clarified in the text. |
| **Figure 9**: I suggest to re-phrase the caption describing what is shown in the figure in more detail. I also suggest to use detailed wording for moment magnitude Mw inferred from y-intercept, hypocentral depth D, and expected epicentral intensity IE (see my | Done. |

| | |
|---|---|
| comment above 2.8.), and to add '... is shown with colored lines for Mw 4.0 (blue), Mw 5.0 (green), and Mw 6.0 (orange)', or the like. | |
| **Line 328-330**:  I recommend to publish the "attenuation curves" of the 206 earthquakes of the analysed set in the supplements. | We added the attenuation curves of the 206 events of the analyzed-set in the supplementary material (supplement S2 .zip file). |
| **Line 333-340**:  There may be pro's and contra's of a step-by-step and a joint-inversion method. Given the well-known trade-off between magnitude and depth, a joint inversion is appropriate from the outset, in my opinion. I see advantages of a joint inversion particularly in cases where magnitude and depth are poorly constrained, and possibly also in cases of "larger magnitude earthquakes" (see below). | We show that our method is independent of magnitude, meaning that the slope of the attenuation curve calculated within 50 km from the epicenter is affected only by earthquake depth, not by earthquake size, we clarified this critical issue by discussing it more thoroughly in the text. In the cases where both magnitude and depth are poorly constrained, i.e., when relying on a few MDPs, the uncertainty will be greater, whichever method is used. |
| **Line 341-380** (Section 3.7):  The procedure of "variable moving windows" dealing with "larger magnitude earthquakes" is hard to comprehend from the text and poorly justified, in my opinion. What does the slope of intensity attenuation measure in such cases?  It would be helpful to also see the "attenuation curves" of these large earthquakes (e.g. in the supplements).  An additional figure could help to explain the procedure, and in particular the geometries and parameters (Re, RJB, etc.). Several questions remain open, for example, why "every fault" (!) is assumed to have a dip angle of 45 degrees (line 348-349). Apparently Mw 6.75 is adopted as a threshold for the "point source approach" in the method. 21 earthquakes out of 206 of the analysed set seem to need a correction in this respect.  Unfortunately there is no learning set for these larger earthquakes.  Moreover, for the larger earthquake sizes epicenter location and hypocenter depth become less relevant. Would it not therefore be better to limit the use of this method to the smaller earthquakes and leave the larger ones to a case-by-case examination? See my comment above in 2.5. | We tried to explain better the approach of the variable moving windows. The slope of intensity attenuation has the same meaning as for the slopes obtained using our standard method. We added in the supplementary material all the 206 attenuation curves of the analysed-set earthquakes. We preferred not to add an additional figure because we were requested to shorten the overall length of the manuscript and we believe that the approach and the meaning of every variable were clearly described in the text. We understand that our assumption of the average dip of 45° for every fault, irrespective of their kinematics and tectonic setting, it's probably an oversimplification, however we point out that: 1) the value was derived from the overall averaging of the dip angles of all the seismogenic sources of the DISS database (that includes faults of every kinematics and history - i.e. newly formed and inherited); 2) we were forced to use an average dip angle because the exact geometry of each individual seismogenic sources of the large magnitude earthquake subset is not known; |

| | |
|---|---|
| | 3) we wanted to follow a method that was easy to reproduce by everyone, so we decided not to use dip angle values derived from our expert judgement.
The Mw 6.75 was chosen as a threshold because the Re for this magnitude is 10 km, equivalent to our standard radius of the moving circular search areas.
We decided not to use a case-by-case examination because we preferred to assign parameters to the source, and use an automatic method (see answer to 2.5).
However, this problem occurs with all methodologies. |
| **Line 357-365**: The 'cumulative effect' for intensity is a serious problem for the entire method. It is, however, not limited to "larger magnitude earthquakes" (this section). Due to its importance it is better to discuss it in a separate section, see my comment above in 2.6. | We devoted a separate section to this issue in the revised version of the manuscript ("Influence of cumulative macroseismic effects on the depth and magnitude estimation"). |
| **Line 366-374**: The example of the 13 January 1915 Marsica earthquake apparently did not show significant differences of slope after correction. Does this finding hold for all (21) "larger magnitude earthquakes" of the analysed set? | Yes, the difference of slope is little for all events. |
| **Line 379**: The formula for correcting IE does not appear to be a mathematical equation, but a computer program assignment. The meaning of S in this formula is not specified. | The formula for IE correction is a geometric law not a program assigment, S is the steepness we specified in the text. |
| **Line 383**: Which parameters are meant by "their parameters"? | Source parameter, we specified in the text. |
| **Line 390-392**: For Equation 6, I suggest to recall that D is hypocentral depth in km and S is attenuation slope in km-1. For the valid range of application it could be referred to Section 3.2. | Done. |
| **Figure 10**: The depth scale is imprecise; I suggest to improve color gradations and subdivisions of depth scale, and to denote 'Depth D (km)'. Legend for magnitude scale Mw could be improved | Following the suggestion of the Reviewer #2, we denoted the bar with the colour scale for the depth of the earthquakes with "Depth (km)". We also changed the colour scale in order to better |

| | |
|---|---|
| such that it does not use a color from the depth scale legend. What is the sequence in which the data are plotted on the map? Seismotectonic information (back arc extension, ..... etc) is not mentioned in the caption, is it relevant for the earthquake data shown in the figure (see also Fig. 1)? | highlight the different depth classes (hopefully the colour scheme we choose will work better). The magnitude scale does not use any colour, as the earthquake magnitudes are scaled to the size of the circles. We improved the caption of the figure describing in more detail the data plotted in the map. We think that the seismotectonic information could be useful for characterizing earthquakes. |
| **Line 394-395**: I suggest to clarify, rephrase the sentence, and clearly distinguish the cases D < 5.0 and D = 5.0, as well as D = 73.0 and D > 73.0 km (also for in Table S1). | The equal case is the limiting case, we believe it is more conservative to leave the classes merged. |
| **Line 397**: "instrumental location of the learning set earthquakes", I suspect this relates to their depth in particular. | We explained it in the next sentence. |
| **Line 404**: "departure", wording? | We rephrased this sentence. |
| **Line 405, 411, etc**: "mean squared" or 'root mean squared'? | It is the standard deviation, we specified in the text. |
| **Line 407-408**: I suggest to add a reference for the "Boxer method". | Done. |
| **Line 415-419**: Which results (Table S1) "may appear unrealistic"? -- If some results seem unrealistic, I would propose that the authors label these results as 'apparently unrealistic', or, if there are serious doubts, even omit them, together with a justification for doing that. | Reviewer #2 is right, the phrase is rather generic and we should have defined on which basis some of the results appear unrealistic, labelling them as such. To address this observation we modified Table S1 adding the confidence intervals for the depth, and consequently also for the magnitude. As a general rule, we can not use a seismotectonic or any other geology wisdom based approach to define a result as unrealistic or not, because we don't know in advance the true values and the geometry of each seismogenic source. However, all the expected depth ≤ 5 km for earthquakes of the size of M 6 or larger, may be considered as "less reliable", since, on the average, the depth of the earthquakes of that size are usually around 8-10 km. |

| | Besides, that class of expected depth may be indicative that the data used suffer from inherent uncertainties that are reflected in the determination of the steepness and of the y-intercept, that may include the cumulation of damage from multiple main shocks, unpredictable anomalies in wave propagation, strong source directivity and site amplification effects, all of which may also cause a sizable shift in the epicentral location.
We still preferred to indicate depth and magnitude even in cases of high uncertainty because we believe that it can still be important information for historical events. |
|---|---|
| **Line 420-423**: This statement (and example) is evident from Equation 5 and Figure 9, and hence fits better in Section 3.6, I suppose. | It seems to us more suitable for the paragraph "Reliability of the magnitude estimation method" because it is useful to better understand that the reliability of the magnitude calculation, depends on the depth since our is a two-step method. |
| **Line 412, 424-442**: Comparing magnitude Mw estimates using the "Boxer method" with those using this method, which are considered more reliable and why? | This is explained in paragraph 3.9 but we can only compare recent earthquakes (Table S2) for non-instrumental earthquakes, we do not know which source parameters are better, we can only say that the two methods give parameters in overall agreement, given the great diversity between the 2 methods. |
| **Line 431-435**: From my point of view, I would not call the two estimates "generally consistent", but rather 'slightly but significantly different'. Is there any idea how this difference can be explained? | The Boxer coefficients for calculating magnitude are also calibrated using pre-1960 and post 1960 earthquakes but macroseismic moment magnitudes calculated using re-evaluated coefficients are, on average, lower than those reported by the CPTI15 by 0.144.
The differences are due to the great diversity of methods. |
| **Line 432-434**: Why do the magnitudes of "pre-instrumental earthquakes" (i.e., no seismogram data available) depend on "differences in the response of pre-1960 seismographs"? I suggest to clarify. | Instrumental data are also present in some cases, we clarified. |

| | |
|---|---|
| **Line 437-440**: "It is important to be aware ...", does this statement refer to the Boxer-Mw or to both? | Refers to both, we specified it. |
| **Figure 12**: Are the data shown in the figure "all the events" of the analysed set (Table S1)? I suggest to clarify in the caption. | Done. |
| **Line 441**: "... on average our seismic moments are 2.3 times larger than those obtained with conventional methods", I suggest to give a reference for the M0(Mw) relation used. What is meant by "conventional methods" and which M0 values have been compared? | The reference is Hanks e Kanamori (1979) that we added in the label of table S1.
By conventional method we mean the boxer method. We have modified the text by specifying. |
| **Line 443-470**: I suggest starting (rather than ending, line 463) the conclusion section with the usefulness of this method for deriving depth and magnitude from macroseismic data of historical earthquakes in Italy. An a priori definition of "pre-instrumental" and "historical" in the context of this paper would be helpful. Without further investigations, all findings should be limited to Italy. | We have rephrased the Conclusion but kept the original structure. The methodology has been tested in Italy with Italian datasets, so we are aware that it can not be taken "as it is" and applied to other countries (having similar macroseismic datasets of historical earthquakes), however we believe that the workflow may be useful also abroad. |
| **Line 447-450**: I suggest mentioning that HSIT data were only used in the learning set due to availability. The statement "HSIT data were .... almost always the only available observations", is not clear at this point. Does "observations" mean "macroseismic observations" here? I suggest to clarify. | Done. |
| **Line 454-455**: '... independent of magnitude up to a certain threshold' ? (see above). | A maximum magnitude of 6.5 was used for the learning set, since there are no higher magnitudes in the examined period. For higher magnitudes, however, we decided to use a modified method, since the point source approach is limited. Therefore, we did not indicate an upper limit in the text. |
| **Line 467**: 'The historical records in Italy ...', I suppose. | Done. |
| **All tables**: I suggest that the table headings and column headings be described and specified in more detail, even if the details have | We followed the suggestion of Reviewer #2, see our answers specific to each single Table. |

| | |
|---|---|
| already been explained in the text. This is especially necessary for the supplements since the text is not in the same file. I recommend that the authors perform a quality control on the tables. I also suggest that the content of all tables be standardized in terms of format, resolution, terminology, layout, alignment, missing decimal places, etc.  For unification of terminology see my comments in 2.8. above. Physical units are needed throughout. | |
| **Table 1**:  I suggest to complete the table caption and the column headings, for example: moment magnitude Mw (my question: are the values in this column all instrumental Mw?), source of Mw (refer to the list of References), epicentral longitude (degree E), epicentral latitude (degree N), hypocentral depth (km), source of depth (Two questions: 1. Depth values in this table are supposedly instrumental, why is depth termed "estimate" at his point?; 2. Is this also the source for the epicenter?), number of MDP's within 55 km epicentral distance, total number of MDP's, data source for MDP's, attenuation slope S (km-1, this study), standard error of the attenuation slope S (km-1, this study) (Questions: How is it calculated? Why "bar", from Fig. 5 one could assume that "bar" denotes the doubled error, this is to be clarified, see comment in 2.3.), intensity intercept value IE (MCS, this study). -- I suggest aligning the numbers in columns appropriately. -- The results of the learning set of this work are shown partly in Table 1 (attenuation slope and intensity intercept), partly in Table S2 (inferred depth and inferred moment magnitude ("intercept Mw")).  Table 1 and Table S2 are mostly identical (or should be). Hence, why not just merge Table S2 with Table 1? | We completed the table caption and the column headings where needed as suggested by the Reviewer #2. The values in the column Mw are all instrumental Mw.

We cancelled the adjective "estimate" to avoid any confusion.

Latitude and longitude source parameters have been instrumentally obtained and derived from CPTI15 v2.0 catalogue, as specified in the figure label.

We also added the column ""Depth uncertainty".

The bar is one standard error below and one standard error above the steepness value (in total twice the standard error).

We prefer not to merge Table 1 and Table S2 to avoid confusion. In Table S2 we recalculate the depth and magnitude with our method for verification purposes, and we do not want confusion between the instrumental parameters of the learning set, which we need as "Rosetta stone" for historical earthquakes, and the estimated ones. |
| | |

| | |
|---|---|
| **Table 2**: I suggest to complete the table caption: 'Comparison of macroseismic Mw estimates (this work versus Boxer-code results) with instrumental Mw for 15 earthquakes of the learning set.', or the like. I suggest also providing ID numbers of the events listed for reference with Table 1; "Time UTC" and "Source of instrumental Mw" columns can then be omitted from the table. A quick spot check revealed that ID 2 dated 2-May-1987 is listed here as 5-May-1987. In addition, some Mw values (y-intercept, this work) differ from those reported in Table S2. I suggest consistency of the data in the tables. I assume that further quality control is needed. -- Taking the instrumental Mw as the reference, the differences with regard to Mw (this work) and Mw (Boxer) can be compiled and evaluated. Is there an improvement in Mw (this work) over Mw (Boxer)? | We checked again, there were small differences due to approximation because in one case the values were calculated manually in another with the specifically created program. It cannot be called an improvement, but a different method. Our methodology also allows us to estimate the depth of the earthquake and takes this into account to estimate the magnitude. However, we can say that the simplicity of our method makes it easily applicable. |
| **Supplement Table S1**: I suggest that the table be explained in more detail in the caption and in the column headings along the lines I have commented for Table 1 (above), particularly because Table S1 is part of the supplement and hence not contained in the paper itself. For this reason, references to the text, equations, figures, reference list of the paper are needed. I suggest to unify terminology, wording, data resolution and format, sequence of columns, etc. as much as possible with that of Table 1 (which could be combined with Table S2, as suggested) and with that of the text of the paper. Seismic moment M0 is derived from the corresponding Mw, I assume (a reference is missing, I suggest to use the Hanks & Kanamori, 1979 formula), and is thus not independent information in the table. -- In particular, I suggest to complete: Physical units (if any), explanation of the data source abbreviations and references to DBMI, CPTI15, epicentral longitude in degrees E, epicentral latitude in degrees N, moment magnitude Mw,  attenuation slope S (km-1), inferred hypocentral depth D (km), number of mobile averages within the first 55 km | We followed the suggestions. We added the 4 columns on the uncertainties estimated by our method; 2 related to depth uncertainties and 2 related to magnitude uncertainties. We specified the citation of Hanks & Kanamori (1979) for the calculation of seismic moment. |

| | |
|---|---|
| epicentral distance (what does "mobile averages" mean?), standard error of attenuation slope S (this column could be expected next to the slope), number of azimuth slices (the meaning of "azimuth slices" is unclear), number of MDP's ..., intensity intercept value IE (MCS), moment magnitude Mw (from y-intercept, this work), seismic moment M0 (Nm, from ... of CPTI15), seismic moment M0 (Nm, from Mw of this work), or the like. | |
| **Supplement Table S2**: The contents of Table 1 and Table S2 largely overlap. I suggest to merge Table S2 with Table 1 (see above). | We preferred not to merge Table 1 and Table S2 (see above). |

---

## Referee Report (RR1)

Report of the review of « Modern earthquakes as a key to understanding those of the past: the intensity attenuation curve speaks about earthquake depth and magnitude »

**General Comments**

The authors present a two-step method to estimate magnitude and depth from macroseismic intensity. This method uses only data wihtin the first 50 km around the epicenter. First step estimates depth from the steepness of the decay of intensity and second step uses the depth estimated in the first step and epicentral intensity to compute magnitude. The method was calibrated and applied to italian data.

In my opinion, the strong points in the paper are the accessibility of the data used and the bibliographic study to collect reliable instrumental depth (finding reliable instrumental depth is generaly challenging itself).

However, even if the authors did a lot of changes since the last review, the paper is still difficult to read. I found also some misuses of the bibliography. For these reasons, I recommend a major revision of the paper.

**Specific comments**

The authors cite Kövesligethy, 1907, Sponheuer, 1960 and Musson, 1996 (lines 291-294) and write that « nearly all the methodologies developed in the past to calculate depth use magnitude as an essential input parameter », which is wrong. Indeed, Kövesligethy, 1907 used a mathematical formulation to estimate depth from the decay of macroseismic intensity with epicentral distance which does not include magnitude. Moreover, Sponheuer did an inventory of existing methodologies to estimate depth from macroseismic intensity including mehtodologies that do not use magnitude, especially the Kövesligethy, 1907 methodology. Nowadays the Kövesligethy, 1907 methodology is refered to as Sponheuer 1960 methdology, as in Musson, 1996. Musson, 1996 modified the Sponheuer methodology to estimate depth, once again without including magnitude in his methodology. Ambraseys, 1985 and Levret et al, 1996 estimated also depth based on Sponheuer methodology, independently from magnitude. In the work of Traversa et al, 2018 and Provost and Scotti, 2020, the magnitude and depth are not used as an input but as an unknowns. The authors should correctly use the bibliography.

Lines 305-307, the authors writes that « Conversly, a functional form containing both magnitude and distance as independent terms would lead to a change in the shape of the attenuation curve with distance and to a variation of the steepness for a variable magnitude ». I don't understand this sentence. What do you mean by independent terms ? As is it done in the Musson, 2013 and the Tosi et al, 2015 IPE ? If it is the case, this sentence is wrong : I did the check for different depths, using for each depth different magnitudes. For each depth, all curves (obtained with different magnitudes) present the same steepness.

*Title*

I agree with the comments of the previous reviewer 2 : the title is too general in the first part and too vague in the second. The title does not help the reader to understand the exact content of the paper. When I read the title, I except a more general approach than that the one described in the paper.

*Introduction*

Introduction is long and quite confusing.

Why did the authors add a part about the half-degrees and decimal intensities in the introduction ? This part should be either deleted or moved with the description of the distance binning method line 159. In this case, the authors should precise if they use integer intensities, half-degree intensities or decimal intensities as « raw » data before using the distance binning method.

The introduction after line 84 should be reorganized to reflect the plan of the paper. This will help readers to find their ways in the paper.

*2 Seismotectonic complexity and depth variability of Italian earhtquake*

This section is quite confusing for the reader. It also introduces the notions for example of « new faults » and « inherited faults », which are not used afterwards. I understand that it is important for the authors to stress out the large variability of the depths in Italy and thus the importance to take into account depth when estimating magnitude. However this could be done in two or three sentences and could be done in the introduction or in the introduction of section 3.

*3 Methodologies and data analysis*

The introduction of the long section 3 should also include a short description of the second step of the proposed methodology, i.e. the magnitude estimate. Currently, it only describes the first step of the method.

Line 159-160 : the part of the sentence « we use only well-located earthquake[…]. » should be moved in the 3.1 Data selection criteria.

*3.1 Data selection criteria*

The authors should explain the two first criteria : why did the authors add criteria on magnitude to select their learning dataset ?

*3.2 Analysis of the learning set*

Line 268-270 : I don't see on the figure the difference of attenuation between the northern and the central-southern Italy datasets after 50 km. The authors should add in the text additional information, as for example the mean steepness after 50 km between the northern and the central-southern Italy datasets.

*3.5 Reliability and validation of the depth estimation method*

The authors should give more details about the last sentence of this part. When I read the title, I expect a comparison between instrumental depth and the depth estimated by the authors. This part is missing. A figure similar to figure S1 or figure 10 in the supplements would be welcome.

*4 Reliability of the magnitude estimation method*

The authors should include their estimated depth in the comparison with CPTI15 in Figure 12, for example as a color of each point. It would perhaps (i) help them to explain the differences observed between the two magnitudes, (ii) highlight the particularity of their two-step methodology.

**Minor comments**

Line 19 : I would not write that depth and focal mechanism are generally well-known. A favorable network geometry around the epicenter is necessary to have reliable depth and focal mechanism, which is rarely the case.

Line 45 : « Itlay affords a unique opportunity to explore this often overlooked problem » : which problem ? The use of integers, half or decimal intensities ?

Line 54 : earthquake instead of eqrthquake

Line 316 : even instead of eve

Line 369 : I would moderate the simpler and more intuitive part of the sentence. From my point of view, it would take more time to use a two step method than using the joint inversion. In my opinion, the part « it may allow a geological verification of the depth before estimating magnitude » is enough to enhance the authors methodology. Indeed, it is important to check at least the depth estimates in the light of geological and known seismicty when computing historical parametric catalogues.

---

## Author Response (AR2)

**Rev. #1**

**General Comments**

| | |
|---|---|
| The authors present a two-step method to estimate magnitude and depth from macroseismic intensity. This method uses only data within the first 50 km around the epicenter. First step estimates depth from the steepness of the decay of intensity and second step uses the depth estimated in the first step and epicentral intensity to compute magnitude. The method was calibrated and applied to Italian data. In my opinion, the strong points in the paper are the accessibility of the data used and the bibliographic study to collect reliable instrumental depth (finding reliable instrumental depth is generally challenging itself). However, even if the authors did a lot of changes since the last review, the paper is still difficult to read. I found also some misuses of the bibliography. For these reasons, I recommend a major revision of the paper. | First of all, we would like to thank anonymous Reviewer #1 for his/her comments, which were helpful in improving the clarity and readability of the text. We have followed as much as we could his/her suggestions in the revised version of the manuscript. |

**Specific comments**

| | |
|---|---|
| The authors cite Kövesligethy, 1907, Sponheuer, 1960 and Musson, 1996 (lines 291-294) and write that « nearly all the methodologies developed in the past to calculate depth use magnitude as an essential input parameter », which is wrong. Indeed, Kövesligethy, 1907 used a mathematical formulation to estimate depth from the decay of macroseismic intensity with epicentral distance which does not include magnitude. Moreover, Sponheuer did an inventory of existing methodologies to estimate depth from macroseismic intensity including methodologies that do not use magnitude, especially the Kövesligethy, 1907 methodology. Nowadays the Kövesligethy, 1907 methodology is refered to as Sponheuer 1960 methodology, as in Musson, 1996. Musson, 1996 modified the Sponheuer methodology to estimate depth, once again without including magnitude in his methodology. Ambraseys, 1985 and Levret et al, 1996 estimated also depth based on Sponheuer methodology, independently from magnitude. In the work of Traversa et al, 2018 and Provost | We thank the reviewer for the clarification. We modified the sentence to read, "Instead, other methodologies (Traversa et al, 2018; Provost and Scotti, 2020) are subject to a trade off between depth and magnitude, as both parameters are treated as unknown. Our approach is similar to that of (Kövesligethy, 1907; Sponheuer 1960; Musson, 1996) which is based on isoseismals drafting, but directly uses the fit of the attenuation curve computed on averages of the original MDP computed inside moving circular windows." |

| | |
|---|---|
| and Scotti, 2020, the magnitude and depth are not used as an input but as an unknowns. The authors should correctly use the bibliography. | |
| **Lines 305-307** The authors writes that « Conversely, a functional form containing both magnitude and distance as independent terms would lead to a change in the shape of the attenuation curve with distance and to a variation of the steepness for a variable magnitude ». I don't understand this sentence. What do you mean by independent terms ? As is it done in the Musson, 2013 and the Tosi et al, 2015 IPE ? If it is the case, this sentence is wrong : I did the check for different depths, using for each depth different magnitudes. For each depth, all curves (obtained with different magnitudes) present the same steepness. | We thank the reviewer for noticing this misspelled sentence. We meant to say quite the opposite, so we rephrased the sentence as follows: "Conversely, a functional form containing a term combining magnitude and distance would lead to a..." |
| **Title** I agree with the comments of the previous reviewer 2 : the title is too general in the first part and too vague in the second. The title does not help the reader to understand the exact content of the paper. When I read the title, I except a more general approach than that the one described in the paper. | We have changed the title following the reviewer's suggestion to: "Inferring the depth and magnitude of pre-instrumental earthquakes from intensity attenuation curves." |
| **Introduction** Introduction is long and quite confusing. Why did the authors add a part about the half-degrees and decimal intensities in the introduction? This part should be either deleted or moved with the description of the distance binning method line 159. In this case, the authors should precise if they use integer intensities, half-degree intensities or decimal intensities as « raw » data before using the distance binning method. | We added this part of the text (lines 31-44) in response to a specific request from the previous Reviewer #1, whom we quote below. "The manuscript needs a comment to qualitative nature of macroseismic intensity and the use of average intensities and rational intensity values instead of integer values". Nevertheless, we moved it to the beginning of the "Methodology and data analysis" section, as requested by current Rev. #1 |
| The introduction after line 84 should be reorganized to reflect the plan of the paper. This will help readers to find their ways in the paper. | We thank the reviewer for pointing this out. We streamlined and shortened the Introduction, which should now be easier to follow and more informative. |
| **2 Seismotectonic complexity and depth variability of Italian earthquake** This section is quite confusing for the reader. It also introduces the notions for example of « new faults » and « inherited faults », which are | We do not see why this section appears "confusing" to the reviewer, and also overly long. We believe that the reason why this section is needed is clearly outlined in its final sentence: |

| | |
|---|---|
| not used afterwards. I understand that it is important for the authors to stress out the large variability of the depths in Italy and thus the importance to take into account depth when estimating magnitude. However this could be done in two or three sentences and could be done in the introduction or in the introduction of section 3. | *"The earthquakes generated by the new faults and by the inherited faults are often geographically overlapped, as seen in the Po Plain (Sbarra et al., 2019a), which makes their seismotectonic interpretation rather difficult if only the epicentral location is available. Conversely, assigning each pre-instrumental earthquake to a specific depth class helps assigning that event to its relevant domain, thus greatly supporting its seismotectonic interpretation and the calculation of accurate global earthquake."*

 Clearly, in this section we describe a seismotectonic occurrence – the strong variability of earthquake depth in Italy – and delineate a fundamental, potential outcome of our work – inferring the depth of historical earthquakes to assign them to the relevant tectonic framework. Plotting historical earthquakes to show how close they fall to existing faults is a common thing to do, but if the depth of those earthquakes is larger than 20 or 30 km their potential association with the surface fault must definitely be reconsidered.

 We necessarily had to postpone the interpretation of our results to a further paper. Nevertheless, we meant to propose a methodology that may be used by other workers in regions that exhibit a similar depth variability, such as Greece or southern Spain. |
| **3 Methodologies and data analysis**
 The introduction of the long section 3 should also include a short description of the second step of the proposed methodology, i.e. the magnitude estimate. Currently, it only describes the first step of the method.

 Line 159-160: the part of the sentence « we use only well-located earthquake[…]. » should be moved in the 3.1 Data selection criteria. | We added a brief description of magnitude calculation at the end of Section 3.

 We moved the sentence at line 159-160, as suggested. |

| | |
|---|---|
| **3.1 Data selection criteria**
The authors should explain the two first criteria: why did the authors add criteria on magnitude to select their learning dataset? | As stated in the text, these two criteria are the same we used in our previous work (Sbarra et al., 2019). The magnitude selection criteria served mainly to avoid considering too many small intensity/magnitude events, which would result in an incorrect fit to the attenuation curve. A reliable fit requires a sufficient number of macroseismic data up to 55 km from the epicenter. |
| **3.2 Analysis of the learning set**
Line 268-270: I don't see on the figure the difference of attenuation between the northern and the central-southern Italy datasets after 50 km. The authors should add in the text additional information, as for example the mean steepness after 50 km between the northern and the centra-Isouthern Italy datasets. | We rephrased the sentence, adding the average steepness values calculated beyond 50 km of epicentral distance for the northern and central-southern Italy learning-set earthquakes (shown in Figures 3 and 4). |
| **3.5 Reliability and validation of the depth estimation method**
The authors should give more details about the last sentence of this part. When I read the title, I expect a comparison between instrumental depth and the depth estimated by the authors. This part is missing. A figure similar to figure S1 or figure 10 in the supplements would be welcome. | We did compare the instrumental depths with those estimated by our method for the learning set earthquakes: the results are shown in Table S2. A citation to the table has also been added to the end of "Reliability and validation of the depth estimation method" paragraph. We also added a new supplementary figure to the new Fig. S1, as requested. |
| **4 Reliability of the magnitude estimation method**
The authors should include their estimated depth in the comparison with CPTI15 in Figure 12, for example as a color of each point. It would perhaps (i) help them to explain the differences observed between the two magnitudes, (ii) highlight the particularity of their two-step methodology. | As suggested, we added the inferred hypocentral depth in Figure 12, differentiating earthquakes into three depth classes :

0-20 km;

21-30 km;

>30 km |
| **Minor comments** | |
| **Line 19:** I would not write that depth and focal mechanism are generally well-known. A favorable network geometry around the epicenter is necessary to have reliable depth and focal mechanism, which is rarely the case. | We specified that we are talking about "damaging earthquakes", i.e. events whose magnitude is generally large enough to grant good quality data. We also removed "well" from "well-known", to avoid appearing too optimistic. |

| | |
|---|---|
| **Line 45:** « Italy affords a unique opportunity to explore this often overlooked problem » : which problem? The use of integers, half or decimal intensities ? | Here we were referring to the problem of determining what information can actually be gleaned from intensity models and how reliable it is. We rephrased the sentence to make this concept stand out clearly. |
| **Line 54:** earthquake instead of eqrthquake | Done. |
| **Line 316**: even instead of eve | Done. |
| **Line 369:** I would moderate the simpler and more intuitive part of the sentence. From my point of view, it would take more time to use a two step method than using the joint inversion. In my opinion, the part « it may allow a geological verification of the depth before estimating magnitude » is enough to enhance the authors methodology. Indeed, it is important to check at least the depth estimates in the light of geological and known seismicity when computing historical parametric catalogues. | Done. |